# Wind shear enhances soil moisture influence on rapid thunderstorm growth

Christopher M. Taylor[1,2 ✉], Cornelia Klein[1], Emma J. Barton[1,2], Sebastian Hahn[3] & Wolfgang Wagner[3]

Convective storms can develop rapidly, creating hazards to local populations through intense precipitation, strong winds and lightning[1]. The large-scale environment in which thunderstorms develop is often well captured in forecast systems, yet predicting where individual storms will initiate remains a fundamental challenge. It is known that differential heating driven by soil moisture (SM) patterns creates atmospheric circulations that favour convective initiation over drier soils[2,3], whereas wind shear between low and mid levels can enhance storm growth[4,5]. Here we show that the most extreme initiations are especially favoured over SM contrasts by means of an interaction with wind shear. Analysing 2.2 million afternoon events across sub-Saharan Africa, we find 68% more initiations classed as extreme given favourable (versus unfavourable) soil conditions, with greatest vertical storm growth occurring where SM-driven circulations oppose the direction of shear-induced cloud displacement. Developing clouds follow the mid-level wind direction and, where this opposes the low-level flow, rainfall is strongly correlated with locally drier soils. Although such shear conditions are particularly common over tropical north Africa, the effect favours negative SM–precipitation feedbacks globally. The combination of SM heterogeneity and wind shear provides a potentially important source of predictability for where deep convection develops, particularly for the most rapidly developing thunderstorms.

According to the World Meteorological Organization[6], convective storms and their associated hazards were responsible for about 30,000 deaths and approximately US $500 billion in economic losses globally in 2010–2019. Advances in storm early-warning systems are reducing death tolls over time but economic losses are mounting owing to increasing exposure. Moreover, as the climate warms, these storms are projected to become more intense[7]. Such storms typically develop during the afternoon, in response to the destabilization of the atmospheric profile from daytime land–atmosphere fluxes of sensible heat (H) and latent heat (LE). Deep convective initiation (CI) can occur on timescales of tens of minutes as a cumulonimbus cloud deepens rapidly[1]. Updraughts within the cloud generate lightning, whereas evaporation of falling precipitation creates strong wind gusts. The resulting locally intense rain rates can trigger flash-flooding, particularly in urban areas with poor drainage systems.

Atmospheric conditions favourable for intense deep convection are typically characterized by thermodynamic instability of low-level air, often measured by convective available potential energy (CAPE), and vertical wind shear—the change of wind speed and direction with altitude, which influences thunderstorm organization and longevity[4,5]. CI itself is frequently tied to low-level convergence zones that provide a lifting mechanism[8,9], such as along air mass boundaries and storm outflows or within synoptic-scale disturbances[10,11], and can also be triggered by land-surface characteristics[12]. Cloud deepening and growth before CI depend not only on the atmospheric instability but also on the structure of the environmental wind field, which governs cloud motion, entrainment and the maintenance of convective updraughts. Strong directional shear, in particular, inhibits CI where initial updraughts fail to reach a minimum width to withstand enhanced entrainment[13,14] but supports subsequent growth by increasing storm-relative inflow of unstable boundary-layer air[15,16]. The latter makes sheared environments conducive to the formation of severe or long-lived organized storms[17,18].

Although present forecasting systems, based on numerical weather prediction and, increasingly, artificial intelligence, can predict the occurrence of favourable atmospheric conditions for convection over a region, it remains a fundamental challenge to predict where on scales of tens of kilometres (hereafter mesoscale) storms will trigger, particularly in the absence of well-known terrain features, such as mountains and coastlines. Land-surface properties, notably vegetation and SM, can influence convection by their influence on the partition of surface insolation into turbulent H and LE. Although fixed surface properties such as cropland, forest and urban areas can be accurately mapped and incorporated into forecasting systems, SM poses a greater challenge as it changes substantially in response to every new rainfall event, particularly in semi-arid regions.

Many studies examining how SM affects CI have focused on vertical profiles of atmospheric temperature and humidity, in particular, how changing the balance between H and LE can affect whether convection will be triggered[19]. In this view, wetter soils can promote CI by moistening the atmosphere and drier soils by warming near-surface air.

[1]UK Centre for Ecology & Hydrology, Wallingford, UK. [2]National Centre for Earth Observation, Wallingford, UK. [3]Department of Geodesy and Geoinformation, Technische Universität Wien, Vienna, Austria. ✉e-mail: cmt@ceh.ac.uk

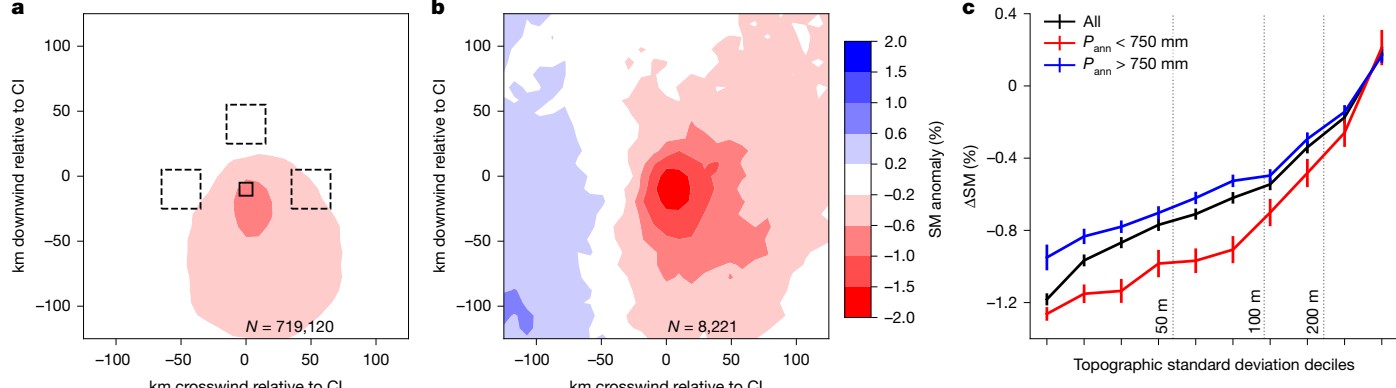

**Fig. 1 | Composite SM patterns preceding afternoon CI events. a,b**, Composite mean SM anomaly patterns (%; shaded) observed before CI events. For each CI event, the data have been rotated around the initiation pixel at 0,0 so that the 100 m wind is flowing bottom to top in the figure, for all cases (**a**) and only cases of extreme convective growth (**b**), corresponding to the 99th centile of local cloud cooling rate. **c**, Local SM contrast ΔSM (%) from all CIs (black), averaged over decile bins of topographic standard deviation (bars denote standard error of the mean). The dataset is also split into climatologically wetter (blue) and drier (red) locations on the basis of a precipitation threshold of 750 mm year⁻¹. ΔSM quantifies the SM difference between the pixel 10 km upstream of the CI (solid box in **a**) and the mean over pixels in the three dashed boxes centred 50 km away.

However, at the mesoscale, observations of CI locations over diverse regions of the world point instead to a primary role for spatial variations in surface fluxes on scales of approximately 10–40 km (refs. 3,20–23). These show that CI is favoured on the downstream edge of drier soil patches. Numerical simulations provide a mechanism for this behaviour—strong contrasts in SM create gradients in H, which in turn drive planetary boundary layer (PBL) temperature and surface pressure gradients and thus mesoscale circulations analogous to sea breezes[24–26]. Convergence associated with these circulations is maximized where the background wind opposes the surface-induced flow[27], hence favouring CI at the downwind edges of dry soils[2,3]. This mechanism explains why, globally, afternoon rainfall is more likely over locally drier soils (a negative spatial soil moisture–precipitation (SM–P) feedback)[28,29]. By contrast, global climate models simulate strong positive SM–P feedbacks because they lack the spatial resolution to capture both SM-induced and convective circulations[28]. In the new generation of high-resolution global convection-permitting simulations, this bias is addressed and a strong, negative spatial SM–P feedback is simulated, alongside a generally weak positive temporal SM–P correlation[30], consistent with observations[29].

Beyond regional analyses in known SM–P 'hotspot' regions[31], it is unclear the extent to which CI is favoured over drier soil more generally, given that the mechanism relies on SM-limited (rather than energy-limited) LE[32], and both high wind speeds[25] and topographic variability[33] disrupt convergence over drier soils. Notably, the effect of wind shear on SM–CI relationships has not been explored, even though shear often plays a key role in severe storm development.

Here we examine a 21-year CI dataset derived from satellite imagery across sub-Saharan Africa (SSA) (south of 25° N) to answer these questions. The domain provides a broad range of surface and atmospheric conditions, including surface aridity, CAPE and wind shear. We adapt an approach developed in previous studies[20] to identify CI events at time $t_0$ on the basis of rapid cooling to at least −40 °C in cloud-top temperature using high-resolution (15 min, about 3–5 km) satellite imagery from the Meteosat Second Generation (MSG) series (2004–2024). This yields 2,234,556 primary CIs during the afternoon (1200–1800 local time (LT); Extended Data Fig. 1). Cloud cooling rate is defined as the change in the local (within 25 km) minimum brightness temperature from $t_0$−30 to $t_0$ + 30 min. To characterize pre-CI spatial heterogeneity, we use surface SM derived from measurements around 0930 LT of the Advanced Scatterometer (ASCAT; 2007–2024). Clear-sky daytime land-surface temperature (LST) imagery (also from MSG) provides an independent source of surface conditions to validate our results, and

continuous lightning data from Meteosat Third Generation (MTG) are available from July 2024. Pre-initiation low-level (100 m) and mid-level (650 hPa) winds are from the 5th European Centre for Medium-Range Weather Forecasts Reanalysis (ERA5)[34], degraded to 1° resolution, with shear defined as the vector difference between winds at these levels, whereas Integrated Multi-satellitE Retrievals for GPM (IMERG)[35] provides precipitation data.

## Composite analysis

Following previous studies[3,20–22], we performed a spatial composite of pre-CI SM, having first rotated the SM field around the initiation location so that all cases shared a common low-level wind direction. Figure 1a depicts a clear elliptical structure with CI favoured at the downwind edge of dry soil, consistent with those previous studies in SM hotspot regions. Our vast dataset allows us to better explore how the local SM contrast around CIs varies under different conditions. We define ΔSM as the difference between the pixel just upstream of the CI and areas 50 km away (Fig. 1). We find negative ΔSM values for all but the most topographically complex locations, qualitatively consistent with idealized numerical modelling[33]. We also see a weaker signal when sampling climatologically wetter parts of SSA, in which we expect SM to have less influence on LE. Note that our analysis excludes densely vegetated tropical forest regions, in which SM retrievals are not possible (Extended Data Fig. 2). There we would expect little sensitivity of LE (and hence convection) to SM outside the driest periods of the year.

We define extreme CI events as those in the top 1% of cloud cooling rates, corresponding to −78 °C h⁻¹ or more. Of these extreme events, 45.6% reach minimum temperatures of −82 °C or colder (about 17 km altitude; Extended Data Fig. 3), equivalent to typical tropical tropopause temperatures[36]. Compared with the all-events composite, extreme CIs exhibit a markedly stronger SM signal (Fig. 1b). They tend to occur under moderate to strong wind shear; 85% of extreme CIs have shear magnitude exceeding 6 m s⁻¹, compared with 49% for all cases. They are strongly favoured over central and tropical north Africa (Extended Data Fig. 1), a global hotspot of lightning and mesoscale convective system activity[37]. They occur preferentially on the northern side of the tropical rain belt (Extended Data Fig. 2), in which wind shear[38] and SM heterogeneity[39] tend to be larger than within the rain belt. However, the larger-magnitude signal in Fig. 1b is not simply because of sampling locations and seasons with stronger space-time SM variability. These cases are statistically more distinct from their background SM states than the all-cases composite (Extended Data Fig. 4).

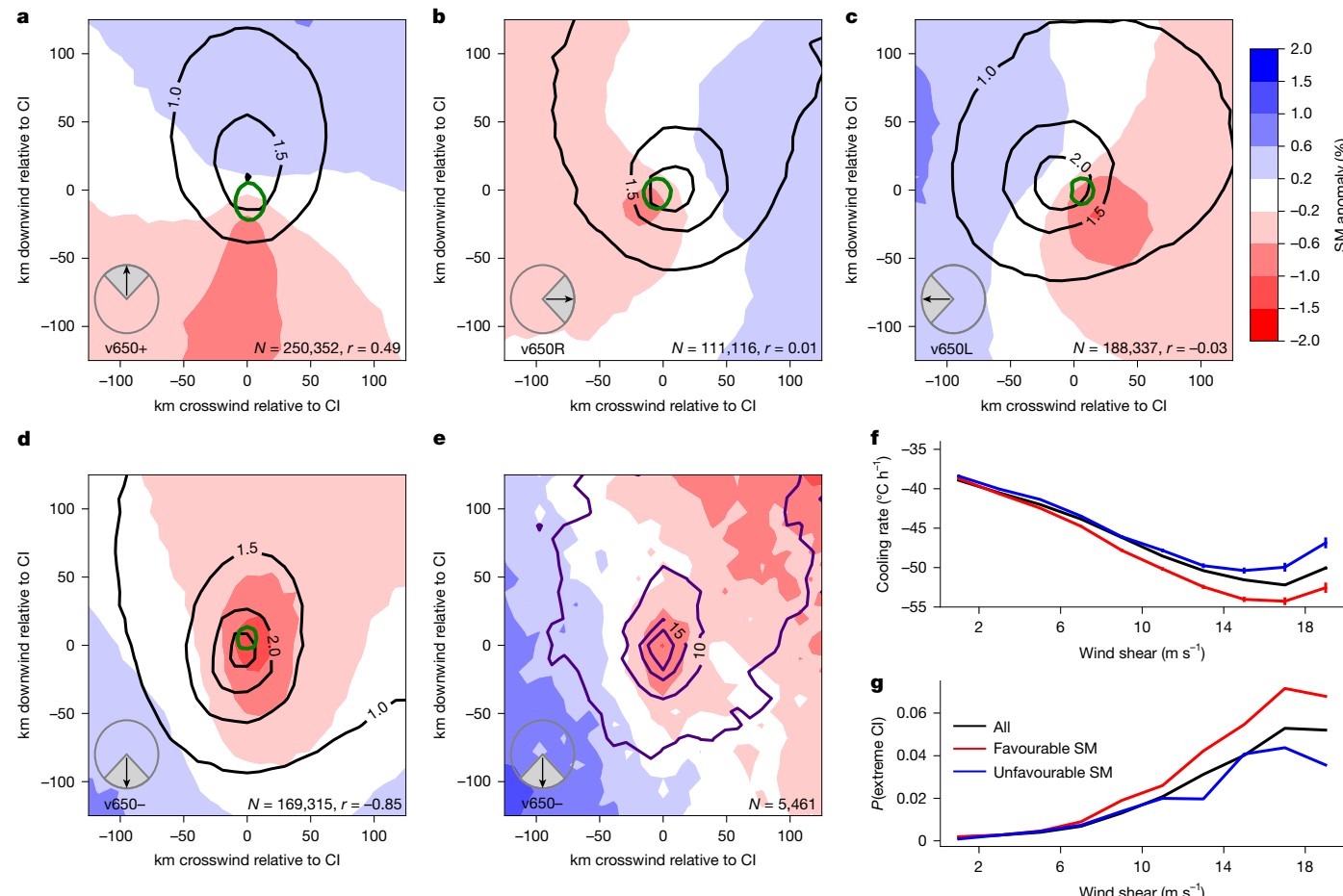

**Fig. 2 | Sensitivity of CI to SM patterns under different shear conditions.**
**a**–**d**, Composite mean SM anomalies (shaded; %) around CI events, rotated according to the 100 m wind, as in Fig. 1. Data are stratified into quadrants of mid-level wind direction (inset) relative to the low-level flow; v650+ (**a**), v650R (**b**), v650L (**c**) and v650− (**d**). Black contours depict 3-h post-initiation P accumulations (mm). The green contour identifies where warm cloud (maximum temperature of 0 °C) cover expands by at least 10% between $t_0$−75 and $t_0$−45 min. The number of events ($N$) and the Pearson correlation coefficient ($r$(SM,P)) are provided for each shear configuration. **e**, Composite mean SM anomalies and lightning activity (indigo contours; flashes over 3 h starting 15 min before initiation) for v650− cases for July–December 2024. **f**,**g**, Cloud-top cooling rate (°C h$^{-1}$) (**f**) and probability of CI being extreme (%) (**g**) as a function of wind shear magnitude for all cases (black) and for favourable (red) and unfavourable (blue) SM patterns for CI. Bars in **f** denote standard errors and cooling-rate differences between favourable and unfavourable patterns are significantly different ($P < 0.001$) for all wind shear bins >2 m s$^{-1}$.

## Sensitivity to wind shear

Figure 2 presents the same set of cases as in Fig. 1a but now stratified according to the direction of the mid-level wind relative to the low-level flow. There is a marked separation in pre-event SM between these four shear directions and, again, the amplitude of the pattern is notably stronger than in Fig. 1a. When the mid-level wind is well aligned with the low levels ('v650+'; 34.8% of all cases), a strong positive downstream SM gradient is the most prominent feature, consistent with arguments underpinning previous analyses[2,3]. However, when the mid-level wind opposes the low levels ('v650−'; 23.5% of cases), the dominant SM gradients are crosswind. Finally, when the mid-level winds blow to the left or right ('v650L' or 'v650R'), they are associated with a rotation in maximum SM gradients of about 45° relative to Fig. 2a, with wetter soils lying downstream of the combined low-level and mid-level wind vector. Taken together, this shows that wind shear direction plays a central role in mediating the influence of the land surface on CI. The all-cases elliptical structure in Fig. 1a is simply a superposition of markedly stronger SM structures from four distinct mid-level orientations.

We now examine how wind shear magnitude influences the SM control on CI. Considering all events irrespective of shear direction (Fig. 2f), cloud growth is strongly sensitive to shear magnitude, with clouds cooling on average by more than 50 °C in an hour for shear exceeding 12 m s$^{-1}$, although there is evidence of a reversal in this trend for shear stronger than 18 m s$^{-1}$. This is consistent with theories about the role shear plays in CI (refs. 13,40). To diagnose the impact of SM gradients on cloud growth, we define direction-specific 'favourable' and 'unfavourable' SM contrasts (ΔSM$_{sh}$) on the basis of composite mean SM structures in Fig. 2a–d (see Methods and Extended Data Fig. 5). Favourable SM configurations (centred on drier soils) accelerate vertical cloud growth compared with the all-cases mean (Fig. 2f), whereas unfavourable patterns (centred on wetter soils) tend to grow more slowly. Moreover, the sensitivity of growth to SM increases with wind shear magnitude. Thus, under high shear conditions, in which extreme CIs are more likely, cloud growth is especially sensitive to SM and this underlies the larger-amplitude composite SM pattern for extreme events in Fig. 1b. For strong shear (>12 m s$^{-1}$), the probability of a CI being classified as extreme (Fig. 2g) is 5.17% for favourable SM configurations, compared with 3.08% for unfavourable SM, a relative increase of 68%.

These shear-based composites also provide distinct behaviour in terms of subsequent rainfall accumulations. Controlled by the mid-level wind, rainfall propagates away from the initiation location. In the case of no directional shear (v650+; 34.8% of all cases), the rainfall

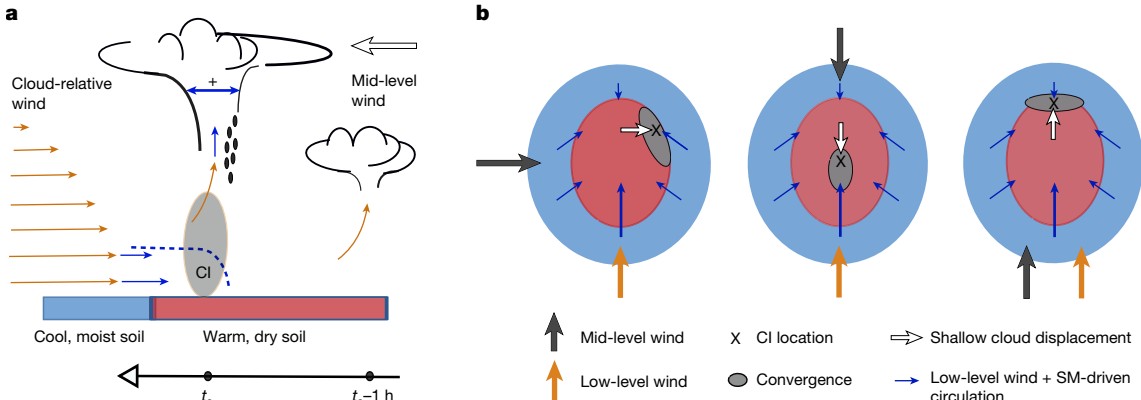

**Fig. 3 | Schematic of SM-affected CI under wind shear. a**, Shallow convection an hour before initiation ($t_0$−1 h) is displaced with the mid-level wind (large arrow), later ($t_0$) encountering strong low-level convergence and ascent (grey shading) where SM-driven mesoscale circulations (blue arrows from moist to dry soils within dashed blue breeze front) act to enhance the cloud-relative wind (orange arrows). This favours the formation of wider, more entrainment-resistant updraughts (blue arrows within cloud at deep CI location). **b**, In the horizontal plane, the favoured location for CI ('x') is displaced from the initial shallow cloud (white arrows) to where convergence opposing the storm motion is maximized (grey ellipses), shown for different shear directions. This zone is created by SM-driven circulations superimposed on the low-level background flow (blue arrows). These configurations favour rapid vertical development along strong SM gradients.

maximum occurs about 20 km downstream, over relatively wet soil. This yields a positive spatial SM−P correlation ($r$(SM,P) = 0.49), consistent with a previous numerical study[2]. In the reverse shear case (v650−; 23.5% of cases), the opposite occurs, with rain highly organized over dry soils ($r$(SM,P) = −0.85) and the amplitude of the composite-mean rainfall pattern markedly increased. For crosswind shear, correlations are close to zero, as rainfall is maximized over the strongest SM gradient in the composite mean. The 6 months of available continuous lightning observations illustrate a highly localized lightning maximum around the CI location linked to SM patterns. Electrical activity is tightly organized within the driest soil pixels in the case of reversed shear (Fig. 2e). For the other shear directions, a well-defined local lightning maximum occurs above the strongest SM gradients (Extended Data Fig. 6).

## Mechanistic interpretation

We propose the following mechanistic arguments to explain our findings. First, we note that, in each of the four shear directions, shallow cloud development at $t_0$−1 h (green contours in Fig. 2) is maximized about 5–20 km upstream of the initiation in the mid-level sense, as expected from a deepening cloud steered in the direction of the mid-level wind. This initial shallow cloud development occurs on the drier side of strong SM gradients for v650+, v650L and v650R and between two opposing strong SM gradients for v650−, locations consistent with idealized model simulations in the absence of directional shear[2]. However, over the next hour as the cloud moves, its growth is expected to be sensitive to changing conditions of instability and convergence in the PBL (Fig. 3a). During sheared displacement with the mid-level flow, cloud-sustaining updraughts must withstand greater entrainment, which inhibits and delays cloud deepening. Only clouds that stay rooted in a favourable PBL, in which wider, more entrainment-resistant updraughts can develop, will continue to grow[13,40].

In this context, our findings suggest that, under wind shear, SM-driven circulations promote particularly strong CIs in locations in which they oppose the mid-level flow, enhancing cloud-relative updraught inflow and growth through low-level convergence (Fig. 3a, $t_0$). For all shear directions, CI ultimately occurs downwind (in a mid-level sense) from initial shallow development. Rapid cloud deepening leading to CI is favoured above strong SM gradients on scales on the order of tens of kilometres, where we expect cloud-relative convergence to peak (grey ellipses in Fig. 3b). In the special case of opposing low-level and mid-level wind, CI instead occurs in the centre of the dry patch, in which

mesoscale circulations from crosswind directions maximize convergent modification of the low-level background flow. This explanation for our satellite observations is consistent with existing observational[41] and model[42,43] studies that note enhanced deep convective development associated with terrain features, mesoscale circulations and surface flux patterns, where resulting larger initial updraught widths can better withstand increased entrainment under sheared conditions[14].

## Geographical variability

Next we examine geographical variability in SM influences on CI and rainfall. Mapped across SSA (Fig. 4a), our ΔSM metric reveals that, outside the mountainous Rift Valley running down eastern Africa, CI consistently occurs over locally drier soil (that is, ΔSM < 0). Over climatologically wetter regions, relatively weak negative ΔSM values arise from a strongly negative dry-season signal balanced by near-neutral wet-season ΔSM (Extended Data Fig. 2), consistent with seasonality in SM controls on LE. Considering the relationship with accumulated rainfall, geographically distinct SM−P feedback regimes emerge when mapping the local spatial correlation between SM and P composites created every 2° (that is, $r$(SM,P); Fig. 4b). Across central and northern tropical Africa, the feedback sign is predominantly negative, whereas in much of eastern and southern Africa, it is positive. Figure 4c,f provides a simple dynamical explanation for these regional differences, which occur despite dominant CI occurrence over drier soils. In southern and eastern Africa, there is only weak directional shear, as quantified by positive values of the cosine similarity of the low-level and mid-level wind vectors ($S_C(\boldsymbol{v}_{low}, \boldsymbol{v}_{mid})$). This implies that CIs there fall largely within either no directional shear (Fig. 2a) or cross-directional shear classes (Fig. 2c,d), yielding generally weak positive SM−P correlations. By contrast, CIs across the remainder of SSA largely exhibit mean negative $S_C(\boldsymbol{v}_{low}, \boldsymbol{v}_{mid})$ values, indicating an important contribution of reversed mid-level winds. This leads to widespread strongly negative spatial SM−P correlations, particularly across tropical north Africa, where thunderstorms develop particularly rapidly (Extended Data Fig. 2).

Finally, on the basis of our findings of a strong relationship between SM−P feedbacks and shear, we reinterpret a previous global observational analysis that examined SM contrasts preceding afternoon rainfall[28]. That study introduced a metric ($\delta_e$) quantifying spatial SM differences between the rainiest and driest pixels compared with a control distribution (see Methods). Effectively, $\delta_e$ and $r$(SM,P) are different measures of the same phenomenon, that is, how SM gradients influence

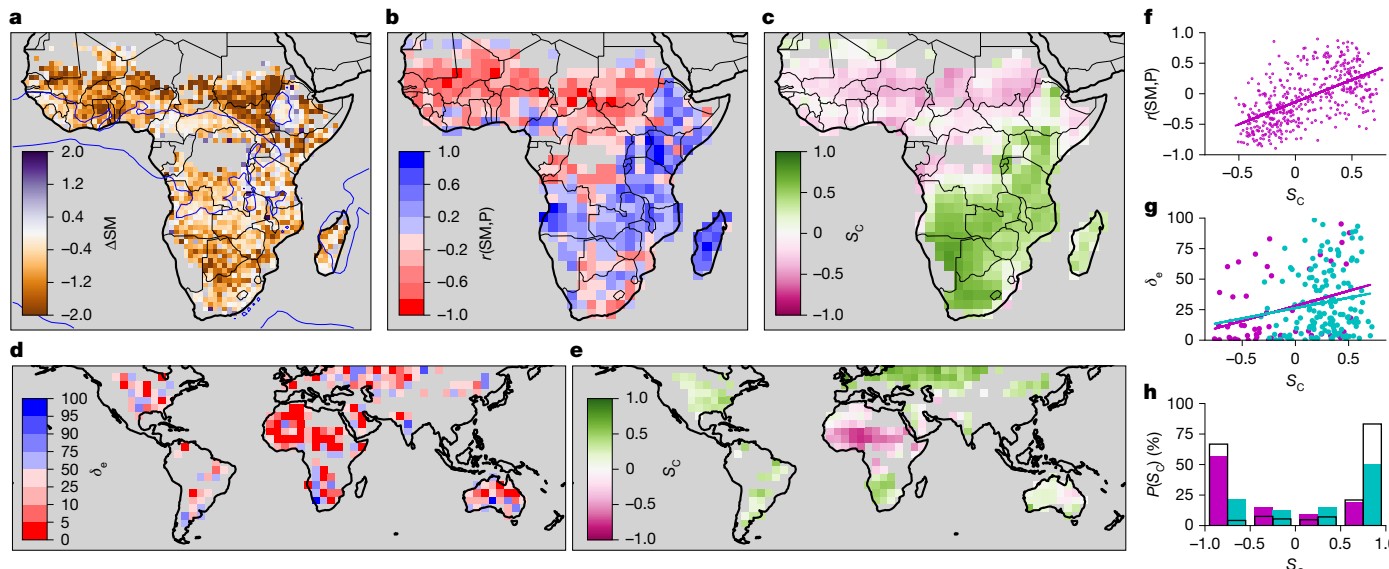

**Fig. 4 | Geographical variability in relationships between CI, P, SM and wind shear. a**, Mean value of pre-CI SM difference ΔSM (%; shaded) and annual mean precipitation of 1,250 mm (blue contour). **b**, Correlation coefficient $r$(SM,P). **c**, Mean cosine similarity between pre-CI low-level and mid-level wind vectors $S_C(\boldsymbol{v}_{low},\boldsymbol{v}_{mid})$. **d**, $\delta_e$ metric from global analysis of location of afternoon rain events relative to pre-CI SM[28]. **e**, Mean $S_C(\boldsymbol{v}_{low},\boldsymbol{v}_{mid})$ between pre-rain wind vectors in **d**.

**f**, Pixel values of $r$(SM,P) and linear best fit as a function of $S_C(\boldsymbol{v}_{low},\boldsymbol{v}_{mid})$ for SSA. **g**, As for **f** but showing $\delta_e$ versus $S_C(\boldsymbol{v}_{low},\boldsymbol{v}_{mid})$ based on the global analysis. **h**, The probability (%) of an afternoon rain event falling within a given $S_C(\boldsymbol{v}_{low},\boldsymbol{v}_{mid})$ bin. In **g** and **h**, data are shown for SSA (magenta) and the rest of the world (cyan). Corresponding probabilities in **h** based on climatological winds are shown as black unfilled bars.

local afternoon rainfall patterns. Low values of $\delta_e$ (signifying rain over drier soils) dominate globally, with tropical north Africa providing the strongest regional-scale signal (Fig. 4d). Considering directional shear, tropical north Africa also stands out, with more positive $S_C(\boldsymbol{v}_{low},\boldsymbol{v}_{mid})$ elsewhere (Fig. 4e). Comparing data from the two maps (Fig. 4g), we find a significant positive correlation over both SSA (consistent with Fig. 4f) and the rest of the world. This indicates that, although directional shear outside SSA tends to be much weaker, shear still plays a mediating role in determining the spatial SM–P feedback. Assuming a global preference for CI over local H maxima, the regional strength of the SM–P feedback will depend on the sensitivity of LE (and therefore H) to SM and the typical lifetime of convection. To this list, we can add wind shear. Notably, afternoon rain days globally tend to occur on days with stronger directional shear (more negative $S_C(\boldsymbol{v}_{low},\boldsymbol{v}_{mid})$) than climatology (Fig. 4h) and implicitly assumed in idealized modelling[2]. So, although our African results occur under globally unusual directional shear conditions climatologically speaking, their implications are relevant for local thunderstorm predictability and SM–P feedbacks in the rest of the world.

## Discussion

Our analysis over a broad range of conditions confirms a dominant role for dynamics in determining where CI occurs within a favourable larger-scale thermodynamic environment. It reveals for the first time to our knowledge the important mediating effect of wind shear on SM–CI relationships, with SM patterns strongly influencing the vertical growth of storm clouds. Evidence for this process is found throughout SSA away from the most extreme topography. In climatologically wet regions, the signal is limited to drier months of the year (Extended Data Fig. 2), when LE is water-limited (rather than energy-limited). We note that our results depend on the realism of the large-scale wind shear in ERA5 over a relatively poorly observed region of the world. Nonetheless, our analysis produces strongly contrasting pre-CI SM patterns when stratified into broad directional shear categories (Fig. 2).

When accompanied by strong directional wind shear, the emerging SM–CI relationship favours strongly negative spatial SM–P feedbacks.

In the absence of directional shear, the SM–P feedback becomes positive, although as illustrated in a previous study[2], the underlying mechanism in this case bears no relationship to widely used thermodynamic arguments[19], even when explicitly considering wind shear[44]. Moreover, although we have focused on SM, other forms of landscape variability can similarly create the thermally induced PBL circulations that influence CI, including irrigated, forested and urbanized areas. The importance of shear in this process suggests a possible explanation for the contrasting responses of convective storms to deforestation in the highly sheared west African environment[45] compared with the weaker shear in Amazonia[46].

The presence of a large-scale heat low circulation in north Africa creates the strongly sheared conditions that set tropical north Africa apart from other areas of SSA. The shear-based mechanism provides a dynamical explanation for why the region exhibits the strongest negative spatial SM–P feedbacks globally. Within this region, observed rainfall in the two days after a storm is favoured over relatively dry areas[39], implying a negative temporal (as well as spatial) SM–P feedback. We suggest that shear-mediated land–atmosphere interactions in topical north Africa contribute to the globally unusual negative temporal SM–P correlation observed there[29], with widespread negative lag-1 daily precipitation autocorrelation[47,48]. The region is also a global hotspot of thunderstorm activity[37]. Although the importance of wind shear on upscale growth of convective systems in this region has been recognized[49], our analysis highlights how the combination of wind shear and SM heterogeneity generates particularly rapid cloud growth in the early stages of storm life cycle. Finally, interactions between poorly observed mesoscale land heterogeneity and CI in the presence of directional shear may also contribute to the notable lack of skill in numerical weather prediction models in this region[50].

Our study has important, globally applicable implications for the predictability of where, given a favourable larger-scale environment, convective storm hazards will develop. The impact of SM patches on CI is markedly stronger when wind shear is accounted for, disproportionately affecting the fastest-growing thunderstorms. When low-level and mid-level winds are in opposing directions, as often found across tropical north Africa, convective activity is greatly enhanced

over drier soil. Favourably aligned with the shear, SM-induced circulations lead to explosive development of deep clouds, with more lightning and larger rainfall accumulations. Such conditions contribute disproportionately to cases for which there is little indication of the imminent storm from satellite imagery even an hour ahead – thunderstorms effectively 'appear out of thin air'. Our results indicate that incorporation of land-surface state information into artificial-intelligence-based or numerical weather prediction models has the potential to improve fine-scale prediction of thunderstorm initiation. This addresses a notable limitation in the provision of early warnings of hazardous storms across the region, which is home to around half a billion people, with an increasing proportion living in urban areas vulnerable to flash-flooding.

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

## Methods

We identify CIs based on the evolution of cloud-top temperature data from the geostationary MSG series of satellites, available from January 2004 onwards. With a temporal resolution of 15 min and spatial resolution of 3–5 km (depending on satellite view angle), MSG provides a critical resource for studying African convection, particularly in the absence of operational radar over most of the continent. We identify cold cloud in the 10.8-micron channel with the widely used threshold of −40 °C. Initiation events are identified when cold cloud appears for the first time ($t_0$) within 30 km, with minimum temperatures within that radius having cooled at least 10 °C h$^{-1}$ over four consecutive time steps. This approach excludes secondary initiations caused by previous convection within 30 km. CIs within 30 km of coastlines or containing at least 10% by area of inland water features (lake, river, permanent wetland according to the Global Lakes and Wetlands Database (Level 3) are also excluded. The geographical location of the initiation is corrected for parallax using climatological profiles from ERA5 to convert cloud-top temperature to height. This results in 88% of cases with offset distances of 10 km or less.

For observations of SM, we use the surface soil moisture (SSM) product derived from the ASCAT flown aboard the Metop satellites from 2007 onwards. For the first time, ASCAT SSM data at a spatial resolution of 15 km (with a sampling distance of 6.25 km) are now available[51]. This new dataset has been derived from ASCAT Level 1b full-resolution backscatter data to achieve the best possible spatial detail. Since the launch of Metop-B in 2014, two or even three ASCAT sensors have been operating in space at the same time, ensuring coverage of nearly all land-surface areas twice daily. The SSM data are provided as both near real-time and climate data record products from the EUMETSAT's Satellite Application Facility on Support to Operational Hydrology and Water Management (H SAF). The retrieval algorithm is based on a semi-empirical change-detection algorithm originally developed for the ERS-1/2 scatterometer missions[52] and represents the degree of saturation (0–100%) in the topmost soil layer (<5 cm). This algorithm has undergone incremental improvements, including an improved modelling of the incidence angle–backscatter relationship[53] and the incorporation of a spatially variable vegetation parameterization[54]. The latest ASCAT SSM data release also includes corrections for long-term land-cover changes that systematically influence backscatter measurements independently of SM variability. Over most of SSA, uncertainty in the retrieval of SSM is less than 4% (Extended Data Fig. 1g). To create SM anomalies from the long-term average for any given month, we compute a climatology based on all observations within that month over the period 2007–2024 but excluding the year in question.

Satellite retrievals of SM can have substantial errors[55], so we use LSTs derived from infrared imagery to validate our analysis. Under clear-sky conditions, LST provides a proxy observation of mesoscale SM variability. In areas in which evapotranspiration is water-limited, drier pixels have higher LST (and lower evaporative fraction) compared with wetter neighbours[56]. As a starting point, we use LST provided by the LandSAF from MSG data at the same spatial and temporal resolution as the cloud-top temperature[57]. We account for the spatial variability in LST owing to differences in vegetation cover, soil properties, topographic height and vegetation root access to groundwater by computing the anomaly in LST compared with a long-term monthly climatology (2004–2015). For every 15-min image between 0700 and 1700 LT, we compare observed LST for that pixel and time of day with its relevant climatological value and compute the daytime-mean land-surface temperature anomaly (LSTA). All LandSAF LST images undergo further screening for cloud based on proximity to cloud-masked pixels and extreme values of the change in LST over successive images compared with the climatology[58]. Note that, although the daytime-mean LSTA can in principle include data after an afternoon initiation, in practice, the persistence of post-initiation cloud effectively excludes sampling post-rainfall LST and the dataset can be considered to represent surface conditions before the initiation.

We use rainfall estimates from version 7 of IMERG, derived from a combination of microwave and thermal infrared imagery, corrected by rain-gauge data[35]. This is available every 30 min on a regular 0.1° grid. Rainfall estimation on the scales studied here is challenging, as high-quality retrievals from microwave are infrequent (typically every few hours for a given location) and the infrared-based interpolation in time and space degrades the quality of the retrievals. We analyse level 2 Lightning Flash data from MTG for the final 6 months of our study period, with data put on a regular 0.05° grid.

We produce spatial composites of surface and atmospheric fields over a domain of 400 × 400 km, centred on the initiation location and oriented relative to the low-level (100 m) wind direction. We use wind data from ERA-5 (ref. 34), downloaded at a resolution of 1° and sampled between 15 and 60 min before the initiation. We prefer 1° resolution to the default (0.25°) to emphasize the larger-scale flow over any local circulation anomalies, for example, related to topography, active convection or SM state, and interpolate the two-dimensional field to the initiation location. The same sampling approach is used for the mid-level wind (650 hPa). We assign each case into one of four directional wind shear classes, based on the angle of the mid-level wind relative to the low-level wind. To focus on mesoscale surface heterogeneity, we remove the domain-mean of the (temporal) anomaly in SM and LSTA before compositing on a regular grid (spacing of 10 km for SM, 5 km for LSTA). As the surface data contain many gaps, we only composite cases for which there are no missing data within 2 pixels of the initiation for SM and within 5 pixels for LSTA. Also for LSTA, at least 25% of the compositing domain must contain valid data. We focus on the 0930 LT ASCAT overpass data in Figs. 1, 2 and 4 but include composites from both that day's LSTA field and ASCAT data from the previous evening (2130 LT) in Extended Data Fig. 7. We use the same approach to create composite maps based on cloud-top temperature (at 5 km) and lightning (at 10 km) (Fig. 2 and Extended Data Fig. 6).

We use several metrics of spatial variability in SM anomalies in the vicinity of each CI. In Figs. 1c and 4a, we compute the difference (ΔSM) in SM anomaly between the pixel 10 km upstream of the CI and the average across boxes (width 30 km) centred 50 km downstream and in both crosswind directions (depicted in Fig. 1a). We chose a distance of 50 km on the basis of previous analysis highlighting SM gradients on length scales of 10–40 km (ref. 3). For Fig. 1c, all events with available SM data are grouped into deciles of the local (within 30 km) standard deviation in topographic height, using the ETOPO1 topography dataset[59] (resolution 1 arcmin; Extended Data Fig. 1). A distribution of ΔSM comprising values for each event is compared in Extended Data Fig. 4 with a distribution of non-event ΔSM values. For each event, non-event samples are computed using the identical box location and day of year but from different years. The curves thus quantify how the probability of CI changes across the climatological distribution of ΔSM. A variant on ΔSM, ΔSM$_{sh}$, is defined using different box locations according to the direction of the mid-level flow relative to the low-level wind. These locations (plotted in Extended Data Fig. 5) are selected to capture the dominant gradients in the composite-mean SM patterns shown in Fig. 2a–d. For each shear direction, events with 'favourable' and 'unfavourable' SM patterns are identified when ΔSM$_{sh}$ falls within the first and tenth deciles, respectively.

As a measure of the influence of SM on local P patterns, we perform a linear regression between composite mean SM and P using all pixels within a 200 × 200-km box ($N = 441$) centred on the initiation. The composites are based on either directional wind shear (Fig. 2a–d) or geographical location (Fig. 4b). For the latter, composite SM and P fields are created by pooling all events that occur within every 2 × 2° box across SSA, irrespective of wind shear, and excluding areas with fewer than 25 events.

In Fig. 4, we place our results on SM–P feedbacks in a wider context by revisiting a previous global study of SM effects on afternoon rain[28]. That analysis was based on considering pre-rain SM contrasts between the locally rainiest ($L_{max}$) and driest ($L_{min}$) pixels in rain events (minimum 3 mm) that develop during the afternoon. In summary, the study used a combination of SM and P datasets gridded at 0.25° over the period 2002–2011 and excluded mountainous and coastal regions. For every event, differences in SM anomalies between $L_{max}$ and $L_{min}$ were computed, alongside a set of control SM differences using the same location and day of year but in different years. The metric $\delta_e$ provides a measure of the likelihood of the difference in the mean SM contrasts (event versus control) and is equivalent to the probability from a two-sided test that afternoon rain is equally likely over locally drier and wetter soils. The analysis pooled events on a 5° × 5° grid between 60° S and 60° N. Here we put the results in the context of directional wind shear by computing pre-event $S_C(\boldsymbol{v}_{low},\boldsymbol{v}_{mid})$ for each event and presenting data in Fig. 4 when there are at least 50 contributing rain events within a 5° grid cell. Given the similar nature of $\delta_e$ and $r$(SM,P), consistent broad-scale patterns in Fig. 4b,d are to be expected. However the scatter plot between $S_C(\boldsymbol{v}_{low},\boldsymbol{v}_{mid})$ and $\delta_e$ for pixels within SSA (Fig. 4g, magenta) is much noisier than for $S_C(\boldsymbol{v}_{low},\boldsymbol{v}_{mid})$ and $r$(SM,P) (Fig. 4f). This is not surprising considering the much smaller sample size in the global analysis, the coarser resolution of its datasets and its reliance on rather uncertain precipitation retrievals to define events (as compared with well-defined CI events based on cloud-top temperatures). To create Fig. 4h, we compare the frequency of directional wind-shear conditions on event days with sampling the monthly climatological winds for each event.

Extended Data Fig. 7 provides a validation of our key results in Figs. 1, 2 and 4 with alternative surface data. We find qualitatively similar composite SM patterns when using ASCAT data from the previous evening (about 2130 LT; Extended Data Fig. 7h–k) as in Fig. 2a–d. This confirms that our results are insensitive to potential rainfall between morning overpass time (0930 LT) and afternoon initiations. Considering LSTA, the all-event composite (panel a) depicts a warm ellipse with maximum values 10–20 km upstream of the initiation, consistent with the SM composite (Fig. 1a). In the case of the LSTA composite however, this is superimposed on a weak, large-scale, positive downwind LST gradient. We find that the large-scale LSTA gradient increases with low-level wind speed. We interpret this large-scale gradient as a response to the large-scale atmospheric temperature gradient, which in turn accelerates the large-scale wind. However, in this study, we are interested in the mesoscale LSTA field and our results (Extended Data Fig. 7) are entirely consistent with the SM analysis. First, the amplitude of the LSTA pattern increases substantially when considering extreme events (Extended Data Fig. 7b, compared with Extended Data Fig. 7a; also found in Fig. 1a,b). Second, the strongest LSTA gradients around the initiation align with the relevant SM gradients when sampling specific shear directions (Extended Data Fig. 7d–g, compare with Fig. 2a–d). Third, ΔLST (computed using the boxes in panel a) exhibits the same functional relationships with topographic variability and climatological rainfall (although of opposite sign) as ΔSM (Extended Data Fig. 7c, compare with Fig. 1c). Finally, the map of ΔLST depicts essentially the same features as that of ΔSM (Extended Data Fig. 7l, compare with Fig. 4a). Prominent differences in the maps arise as a result of a reduced sensitivity of LST to SM in densely vegetated (aerodynamically rough) areas[56] and masking of ASCAT data over desert areas in which sub-surface backscattering effects are important[60]. Despite differences in sensitivity to SM and sampling from the two instruments, the two are moderately well correlated ($r = -0.52$).

## Data availability

Archives containing the raw data used here are available from: https://user.eumetsat.int/catalogue/EO:EUM:DAT:MSG:HRSEVIRI (cloud-top temperature), https://navigator.eumetsat.int/product/EO:EUM:DAT:0691 (lightning), https://hsaf.meteoam.it/Products/ProductsList?type=soil_moisture (SM), https://datalsasaf.lsasvcs.ipma.pt/PRODUCTS/MSG/MLST/ (LST) and https://disc.gsfc.nasa.gov/datasets?keywords=gpm%20imerg%2007 (precipitation). Near-real-time convection and land-surface conditions over Africa can be visualized at https://africa-hydrology.ceh.ac.uk/nowcasting/. Coastlines and borders in Fig. 4 were generated using the Python packages matplotlib[61] and cartopy, made with Natural Earth data – Free vector and raster map data, https://www.naturalearthdata.com/.

## Code availability

Code for identifying and analysing CIs and the full CI dataset are available at https://doi.org/10.5281/zenodo.17871500 (ref. 62).

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

**Acknowledgements** The research presented here was supported by the UK Natural Environment Research Council (NE/X006247/1, NE/W001888/1, NE/X017419/1, NE/X018520/1) and the Met Office on behalf of the Foreign, Commonwealth & Development Office under the WISER Early Warnings for Southern Africa (EWSA) project. We thank the participants of the WISER-EWSA test beds for related discussions. We gratefully acknowledge the providers of the key datasets used here (EUMETSAT, LandSAF, HSAF, the European Centre for Medium-Range Weather Forecasts, National Aeronautics and Space Administration).

**Author contributions** C.M.T., C.K. and E.J.B. conceived the study and discussed the results. C.M.T. performed the analysis and C.K. developed the mechanistic understanding. C.M.T. and C.K. wrote the paper. S.H. and W.W. developed the SM dataset. All authors commented on the paper.

**Competing interests** The authors declare no competing interests.

**Additional information**
**Correspondence and requests for materials** should be addressed to Christopher M. Taylor.

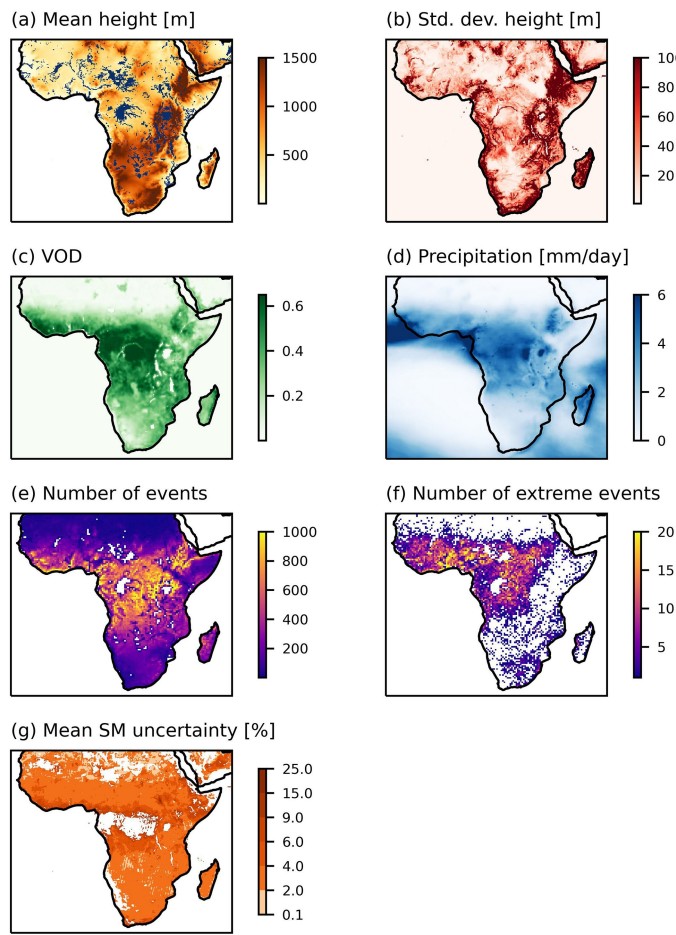

**Extended Data Fig. 1 | Variability across SSA.** Mean (**a**) and standard deviation (**b**) of topographic height (m) at a resolution of 0.25° (derived from https://www.ngdc.noaa.gov/mgg/global/relief/ETOPO1/tiled/). Blue shading in **a** denotes pixels with at least 10% coverage by water bodies (https://www.worldwildlife.org/pages/global-lakes-and-wetlands-database). Climatological mean vegetation optical depth (VOD)[63] (**c**; unitless) and precipitation (**d**; mm day$^{-1}$). Total number of CI events (**e**) and extreme CI events (**f**) per 0.5° pixel over the 21-year study period. Mean uncertainty in SM retrieval (**g**; %), masked out in locations with fewer than 3,000 valid morning observations.

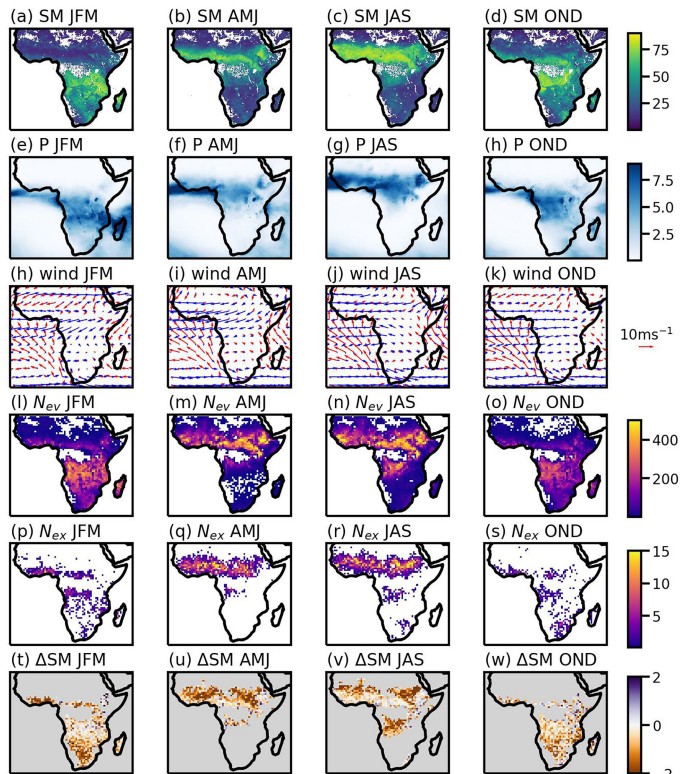

**Extended Data Fig. 2 | Seasonal cycle.** Maps of climatological SM (**a**–**d**; %), precipitation (**e**–**h**; mm day$^{-1}$), wind vectors at 100 m (red) and 650 hPa (blue) (**h**–**k**; m s$^{-1}$), number of CI events used in SM analysis (**l**–**o**), number of extreme CI events (**p**–**s**) and ΔSM (**t**–**w**) for January–March (JFM), April–June (AMJ), July–September (JAS) and October–December (OND).

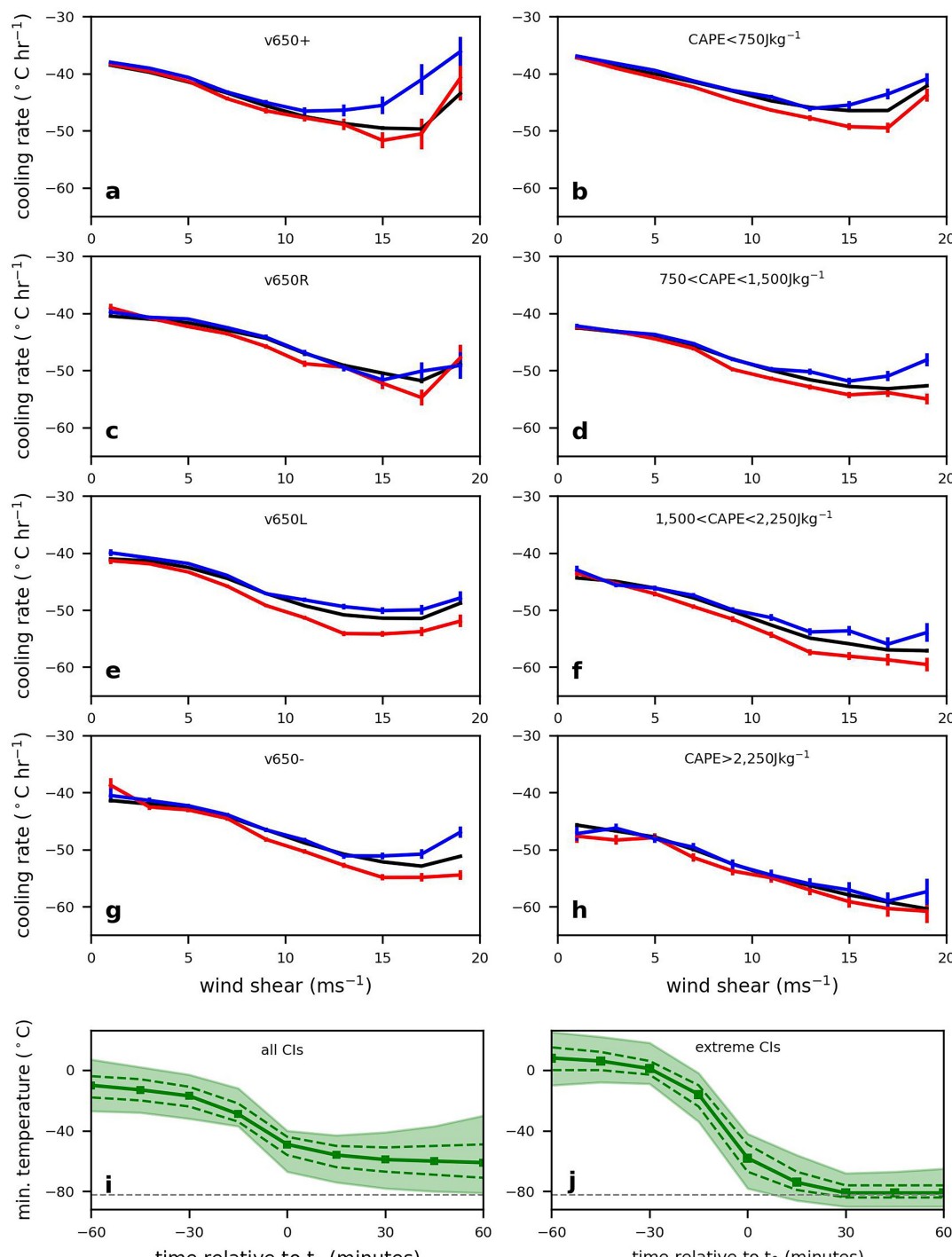

**Extended Data Fig. 3 | Local cloud cooling rate sensitivity to shear direction and CAPE.** Mean local cloud cooling rate (°C h$^{-1}$) from $t_0$−30 to $t_0$+30 min as a function of wind shear magnitude (as in Fig. 2f) for different shear directions (**a**,**c**,**e**,**g**) and different CAPE values (**b**,**d**,**f**,**h**). All SM patterns are shown in black, favourable SM patterns in red and unfavourable SM patterns in blue. Bars denote standard errors. Evolution of local cloud-top temperature minimum (°C) averaged over all (**i**) and extreme (**j**) CI events. Solid line denotes the median, dashed lines are the 25th and 75th centiles and the shading spans the 5th to 95th centile local minimum temperatures. The dashed grey line at −82 °C indicates the typical mean tropopause temperature.

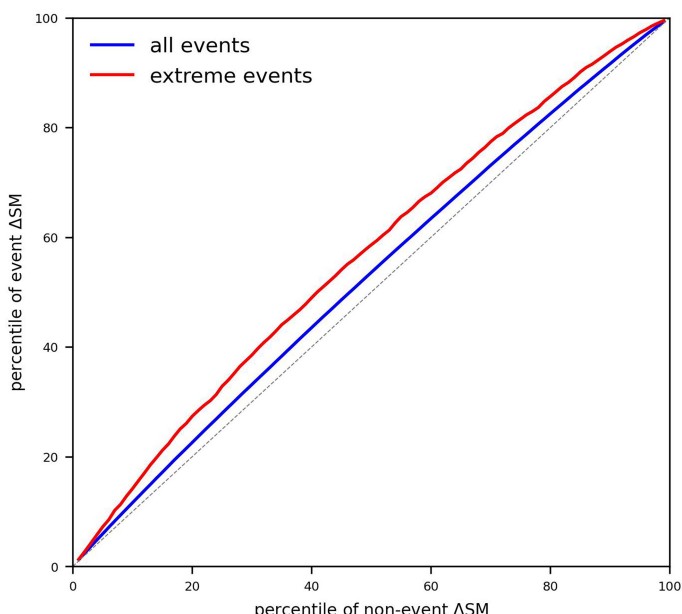

**Extended Data Fig. 4 | Distribution of pre-event SM differences compared with a non-event distribution.** Percentile of ΔSM values for CI events relative to the percentiles from a control distribution of ΔSM for non-CI events. The blue line uses all 474,923 CI events for which ΔSM can be computed, whereas the red line is based on only the 5,357 extreme events, defined as being within the top 1% of local cloud-top cooling rate. The control distributions are constructed from computing ΔSM at the same location and day of year as the relevant event distribution but in a different year from the event. Considering both all and just extreme cases, the event and non-event distributions are different according to a Kolmogorov–Smirnov test at the 99.999% level.

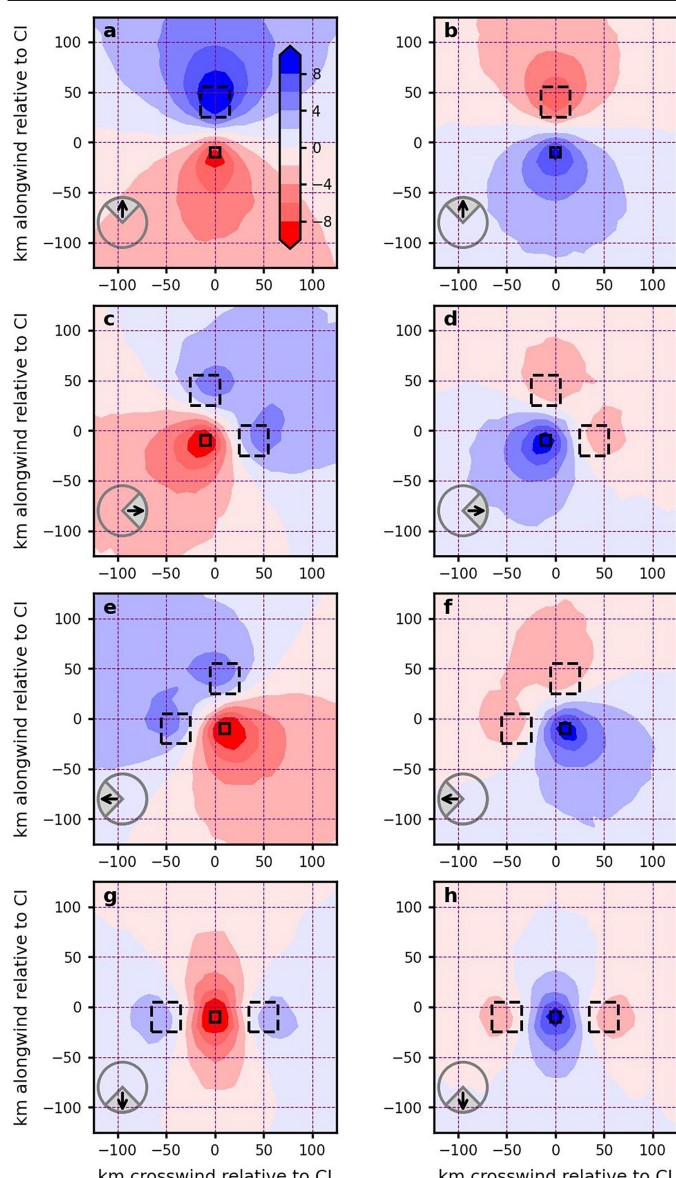

**Extended Data Fig. 5 | Composite mean SM patterns for favourable and unfavourable events.** Composite mean pre-storm SM anomalies (%) relative to CI location created by selecting events within the 1st (favourable; **a**,**c**,**e**,**g**) and 10th (unfavourable; **b**,**d**,**f**,**h**) deciles of $\Delta SM_{sh}$, based on differences in SM between the central (solid) box and the mean over the nearby box(es; dashed). Arrows depict the direction of the mid-level wind relative to the low-level flow.

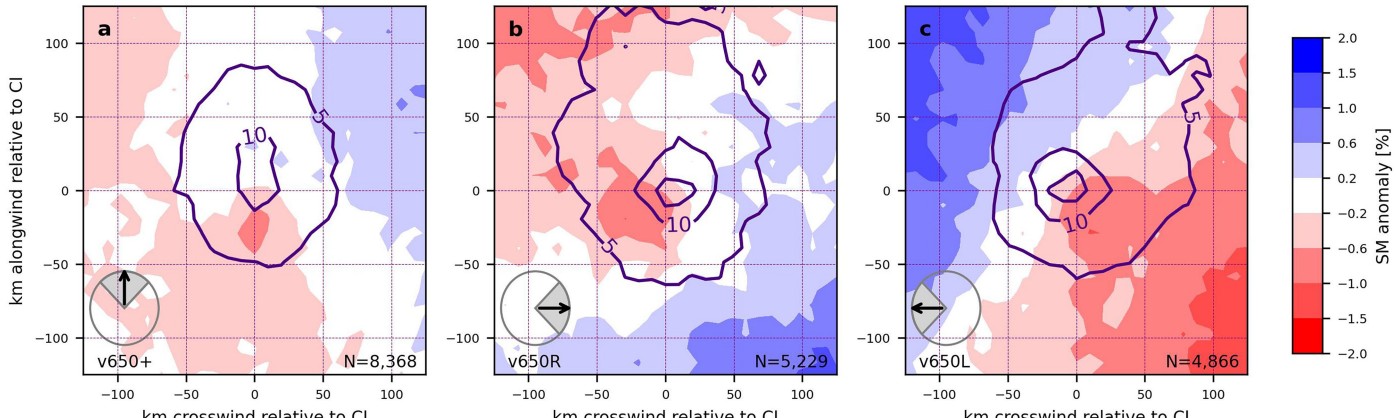

**Extended Data Fig. 6 | Composite mean lightning flashes for three directional shear configurations.** Composite mean lightning flashes (contours) over 3 h starting 15 min before initiation, based on events during July–December 2024. Data are stratified into quadrants of mid-level wind direction (inset) relative to the low-level flow for v650+ (**a**), v650R (**b**) and v650L (**c**). Shading denotes composite mean SM anomalies (%) for these events. The case v650– is shown in Fig. 2e.

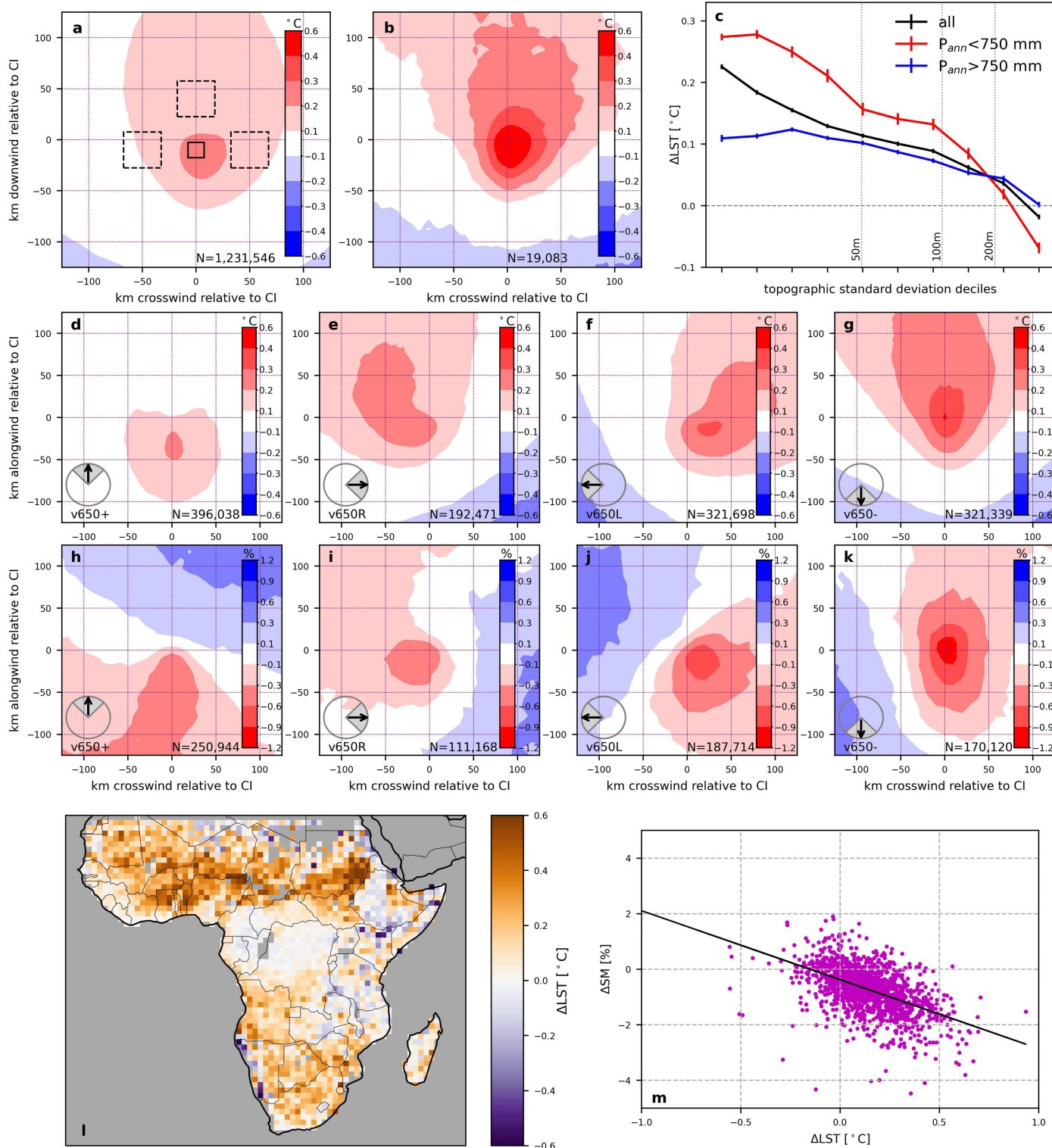

**Extended Data Fig. 7 | Analysis with LST and SM observations from the preceding evening.** Composite pre-storm LSTAs (°C) relative to CI location, rotated according to the 100 m wind (as in Fig. 1), for all cases (**a**), extreme convective growth cases (**b**) and cases for which the mid-level wind is aligned (**d**), towards the right (**e**), towards the left (**f**) and reversed (**g**) relative to the low-level wind (as in Fig. 2). **c**, Local LST contrast (ΔLST; °C) from all CIs (black), averaged over decile bins of topographic standard deviation (bars denote standard error of the mean). The dataset is also split into climatologically wetter (blue) and drier (red) locations based on a precipitation threshold of 750 mm year⁻¹. **h**–**k**, As for **d**–**h** but using SM anomalies (%) observed the previous evening rather than LSTAs on the day of the event. **l**, Mean value of ΔLST (°C) across SSA. **m**, Relationship between pixel values in maps of ΔLST (**l**) and ΔSM (Fig. 4a), with a linear regression line ($r^2 = 0.27$).