## [Peer Review file · Nature]

Wind shear enhances soil moisture influence on rapid thunderstorm growth

Corresponding Author: Dr Christopher Taylor

Version 0:

Reviewer comments:

Referee #1

(Remarks to the Author)

This paper identifies a huge number (2.3millions) of deep convective initiation (CI) events over most of Sub-Saharan Africa (with the exception of very wet regions/seasons) and creates composites of Soil Moisture (SM) and rainfall (P) that are stratified by the background directional wind shear.

They show that

i. inhomogeneous SM patterns are associated with the CI events composites for all cases (away from topography) and reach maximum SM dry/wet differences of about 1% (for context most of the CI events, and in particular the extreme ones are occurring in places/seasons when the background SM is of the order of 50%, at least that's how I see Ex.Fig 2)

ii. For optimal SM inhomogeneity, Extreme CI events (whose cloud deepening is characterized by the fastest cooling rates) double (for the highest shear cases) from 3% to 6%, compared to the least favorable SM inhomogeneity (10th to 90th percentile). On average the difference in cooling rate between favorable and unfavorable SM is small for weak shear, but gets to 5-10°C/hr at high shear, equivalent to the effect of doubling CAPE from about 1000 to about 2000J/kg.

iii. The precursor warm cloud and the rainfall associated with the composite CI events are placed at the edge of the dry soil patch for most cases, but at the center of the dry patch when the low level and upper level winds have opposite direction.

iv. (iii) implies that the spatial correlation between rainfall and soil moisture in a 2°x2° box around the CI is positive for most cases, but negative for the strongest directional shear.

v. Across the continent, CI occurs very consistently over dry patches, but the wind has opposite direction between the upper and lower levels in a minority of cases, mostly confined to Tropical North Africa (mean cosine similarity gets to -.5 there, according to Fig 4c); this implies that the negative spatial correlation between SM and P at the time of CI is mostly limited to the latter region (Sahel+Gulf of Guinea).

All of this is done very convincingly, with great care, with the T's crossed and the I's dotted. The paper is well written and well illustrated. It is a very solid and very interesting piece of science. What I am not completely sure that I understand are the significance statements made by the authors. Specifically, there are two things I am confused about: how this helps with predictability, and what this means about land/atmosphere feedbacks more broadly.

I see how these results could be a source of predictability for initiation at least in a statistical sense: if one had the ability of processing the SM signals to find the most favorable patterns, one could warn of enhanced probability of extreme CI events, distinguishing between locations 100km apart. What I am not so sure about is whether that would also translate in the ability to forecast beyond the initiation, to the development of the MCS that typically are the systems that are the most impactful. (This is of course leaving aside the issue of how a nowcast of this sort would be not just produced, but also disseminated...

that's for the social scientists!) There is a conclusion here (lines 317-318) that the more rapid initiation favors rainier and more long lived storms, but it's not self evident to me and there is no citation for the statement.

I am also wondering whether predictability can be determined by compositing on the CI. Once the CI is identified, one can see that it had a warm cloud precursor right at the downwind edge of the dry patch, where the gradients are strongest (or at the core of the dry patch, for the v850- case). But don't we need to know the complementary: given a dry patch, how likely it is that a CI will occur? It might be that once the shear and the CAPE are there, we already have a good handle on whether CI will occur, and the land information can add some detail... but is this the case? Or is the problem with the forecast skill such that we don't know IF CI will occur, let alone where exactly?

The other issue I am confused about is how we go from the spatial correlation between SM/P for a box around the CI event, to the characterization of the SM/P feedbacks more broadly. My understanding is that the negative spatial correlation might lead to homogenization of the SM field and less likely CI (and vice-versa: a the positive SM/P correlation would be sharpening SM gradients and lead to more likely CI). But how does this translate to temporal SM/P feedbacks, when a lot of the rain comes from remotely triggered systems? On a related note, the connection to the negative autocorrelation of rainfall (at day 1 lag) also seems tenuous, given that the latter seems to be explained by atmospheric waves, and not land processes.

When I say I am confused, it's not a white lie. I genuinely wonder whether I am missing something here that somebody with a deeper knowledge of the literature would be able to see and that would make them come to the same conclusions as the authors. But at the very least these issues should be clarified and discussed less obliquely.

Other minor comments:

L 117: Extended figure out of order.

L 119: a couple of words on how you define the shear and a reference to Methods would be nice

L209: I am a bit unsure of what you mean here by favourable. Are you making a general statement, or are you referring to your favourable/unfavourable SM patterns? Are you suggesting that the updraft is moving from t0-1 hour to t0 to a more favourable environment?

L221: how are your result consistent with the sensitivity to updraft width? What can you say about updraft width?

L245: I would add a couple of words (and a reference to Methods) to indicate that you composite over all CI events occurring in 2°x2° boxes.

L255-256: for clarity, I would say "negative SM-P spatial correlations during CI events, particularly..."

L284: Can you add a reference for this? It makes sense, but I did not know where to read about the effect of typical lifetime convection

L286: globally, rain occurs more often then expected from climatology for the most negative cosine similarity, but the same figure shows the opposite for SSA. Am I missing something?

L317-318: Why rainier and more long-lived?

(Remarks on code availability)

Referee #2

(Remarks to the Author)

COMMENTS TO THE AUTHOR(S)

The manuscript "Wind shear enhances soil moisture control on rapid thunderstorm growth" presents a novel, data-rich, and physically consistent analysis of how wind shear modulates the impact of soil moisture heterogeneity on convective initiation (CI) across sub-Saharan Africa. Using an unprecedented dataset (~2.3 million CI events) and integrating high-resolution satellite observations with ERA5 reanalysis and IMERG precipitation products, the authors convincingly show that the most explosive storm initiations preferentially occur over locally dry soils when wind shear is strong. They further demonstrate that the direction of shear determines the sign and magnitude of soil moisture-precipitation (SM-P) feedbacks.

This study represents a major advance in understanding where and why thunderstorms develop under similar large-scale conditions, providing a mechanistic explanation for negative SM-P feedbacks observed in the tropics. Its implications extend globally, with clear relevance for convective predictability and the evaluation of convection-permitting climate models. The analysis is methodologically rigorous, grounded in theory and observations, and clearly written.

Overall, this is an excellent and highly original contribution. It convincingly demonstrates how mesoscale soil moisture contrasts interact with vertical wind shear to modulate deep convection and offers a robust, physically grounded explanation for the observed spatial SM-P relationships. I consider this paper suitable for publication in Nature pending minor revisions to improve clarity and discussion.

Major Comments

1. The definition of "favourable" and "unfavourable" SM configurations (dSMsh) could be expanded. While the manuscript references Extended Data Fig. 6, it would help to explicitly define how thresholds were chosen.
2. It would be valuable to provide an uncertainty estimate or sensitivity analysis for SM retrieval errors and their impact on dSM values, especially given known retrieval challenges over dry, sparsely vegetated regions.
3. Clarify how parallax corrections were handled quantitatively (e.g., typical offset distances).
4. The discussion would benefit from a more explicit contrast between spatial and temporal SM-P feedbacks. The

manuscript notes that spatial feedbacks are strongly negative under shear, but the implications for daily or subseasonal memory of rainfall could be elaborated.

5. The final discussion paragraph mentions implications for nowcasting and AI-based systems. This is an excellent point. Consider expanding with one or two sentences on how these findings might be operationalized (e.g., coupling near-real-time SM retrievals with convection-permitting nowcasts).

6. The study could briefly acknowledge the absence of direct in-situ flux or sounding data to confirm the proposed mechanisms (entrainment resistance, convergence strength).

7. It would be helpful to note that ASCAT SM measures only the top few centimeters of soil—thus the coupling may differ under conditions of strong vegetation control or sub-surface moisture gradients.

Minor Comments

1. In the Introduction, when mentioning afternoon storm development, cite relevant diurnal cycle studies.

2. In the Results, clarify that “extreme” refers to rapidly cooling cloud tops rather than the most intense storms by rainfall or lightning.

3. In Figure 3, define timing conventions (e.g., t_0 , t_0-1h) within the caption for readers less familiar with CI detection.

4. Ensure consistent use of abbreviations (e.g., “CI” vs “CIs”).

Recommendation

Minor revision: This is an exceptional and timely paper offering a significant conceptual advance in land–atmosphere coupling and convective dynamics. The suggested clarifications would further strengthen its clarity and accessibility to a broad readership.

(Remarks on code availability)

Referee #3

(Remarks to the Author)

This is a review of the manuscript titled, “Wind shear enhances soil moisture control on rapid thunderstorm growth”, submitted to Nature by Taylor et al. The manuscript details the authors’ analysis of vertical (100m- 650 hPa) directional wind shear and convective storm initiation relationships conditioned on mesoscale surface soil moisture gradients. The analysis focuses on sub-Saharan Africa over a 21-year record using: Meteosat Second Generation land surface temperature, cloud top temperature, lightning data; Scatterometer surface soil moisture estimates; NASA IMERG satellite-based precipitation, and ERA5 100m and 650 hPa reanalysis winds. The authors find that the vertical directional wind shear can partly explain the difference in sign and magnitude of previously-reported soil moisture-precipitation correlations that did not consider vertical directional wind shear. In particular, focusing on the most extreme 1% of storms, “...the greatest vertical storm growth occurring where soil moisture-driven circulations oppose the direction of shear-induced cloud displacement. Developing clouds follow the mid-level wind direction, and where this opposes the low-level flow, rainfall is strongly correlated with locally drier soils.” These findings are significant and novel enough to warrant reporting in Nature, and moreover, the analysis is an exemplary application of multiple geostationary and low-earth orbit datasets. However, there are many minor, but important issues that should be addressed prior to publication. These are outlined below in order of line number.

Overall, 1) all figure captions need to be revised for clarity, 2) Figure 3b’s schematic should be improved, 3) the added-value of Fig. 4d-h and related analyses to the discussion/findings should be re-evaluated. If Fig. 4d-h are truly important and central to the authors’ message, then the text, discussion, and figures should be reworked to elevate these findings accordingly.

Ln1/Title: The title should be revised to better represent the actual scope of work. For example, “Vertical wind shear enhances soil moisture affects on daytime convective initiation in sub-Saharan Africa” or “Vertical wind shear affects the role of regional soil moisture gradients on daytime convective initiation in sub-Saharan Africa”, or similar. The terms “control” and “rapid thunderstorm growth” are unsupported by the analysis.

the Abstract should clarify the type of wind shear (i.e., wind speed, direction, or both, with height)

Ln17 re: “sensitive...via an interaction”. The terms “sensitive” and “interaction” suggests a modeling analysis was conducted to perform attribution, which was not.

Ln25 re: “source of predictability”. Predictability is defined by modeling, not through observation. This analysis did not show “soil moisture heterogeneity and wind shear provides an important source of predictability...”

Ln26 how well does ERA5 represent thunderstorms? Are all convective initiation events assumed to be thunderstorms? In reality, not all CIs result in thunderstorms.

Ln 28 suggest “..storms and [their associated] hazards...”

Ln31 re: “increasing exposure”, exposure to what?

Ln35-36 suggest truncation: “Deep convective initiation (henceforth CI) can occur on time scales of tens of minutes (REFERENCE). Updrafts within the cloud...” Here and throughout, the term “updraft” should be used instead of “updraught”.

Ln38 suggest “particularly in [impervious cover-dominated] urban areas with [rapid, higher-volume runoff (REFERENCE)]”

Ln40 “low-level” not “near-surface”; near-surface is 2-10m AGL; CAPE is not near-surface. Often, MUCAPE is used, which is lowest 300 hPa of the atmosphere.

Ln47 “[PBL-top] entrainment”

Ln57 reword “fixed triggers”. Perhaps, orographic or terrain?

Ln58 suggest "vegetation [water content] and soil moisture"

Ln59 suggest "Whilst [land cover classes] can be accurately..." Is it true that cropland and irrigation can be accurately mapped in present-day forecasting systems?

Ln62 suggest "semi-arid [regions]."

Ln 64 suggest "how [the changing] balance" or just "how evaporative fraction"

Ln76,81 was it intentional to cite ref no. 29 in both cases?

Ln82 glaring omission of Koster (2004, Science) reference for "hotspots"

Ln86 wind shear-CI relationship is classical topic of exploration. So, what is relatively novel is the addition of satellite-based soil moisture heterogeneity. Important to rephrase accordingly.

Ln86-87 for truly "severe" storms, driven by synoptic circulations, it is unusual to claim that soil moisture is the "control". Most SM-P correlation occurs in warm/dry season with relatively less intense storms.

Ln88 the precise study period (i.e., 2007-2024 or 2004-2024) should be specified in parentheses. Relatedly, the available MSG data only appears to support an 18-year analysis (see Ln 367).

Ln88 reconsider use of term "imagery"

Ln94 suggest "primary" will need to be defined in this context

Ln96 suggest "spatial [SM] heterogeneity"

Ln100 suggest "[resampled] to 1deg resolution"

Ln100 reference to study that verifies/evaluates 100-m ERA5 winds in sub-Saharan Africa would be appropriate here

Ln103 suggest "composite of [pre-CI] SM where"

Ln104 rephrase. To "align with the low-level wind" of what? Seems to me that both SM and wind fields are rotated so that the prevailing wind becomes southerly or towards the top of the page in Fig.1.

Ln106 re: "consistent with those previous studies". What is important to note is whether the findings are consistent with studies over the shared study domain (i.e., sub-Saharan Africa and Sahel) and include those references.

Ln107 the analysis does not "explore sensitivity to controlling parameters of the local SM gradient" because it is not a modeling study

Ln109 suggest "most topographically-complex locations"

Ln 110 in the era of AI/ML and hybrid physical models, it is important to clarify the type of idealized modeling. Are these regional climate modeling?

Ln suggest "of [sub-Saharan] Africa" consistent with study domain

Ln115 suggest "events as[those in the top 1%] of cloud cooling rates"

Ln119 suggest "strong [vertical] wind shear"

Ln123 suggest "[tropical] rain belt" or more descriptive reference

Ln125 re: amplitude. Rephrase. Fig 1b does not show a wave.

Ln148,149,187, 255, 401 here and elsewhere, suggest replacing "steering" with "mid-level" for consistency and correctness. 500 hPa is considered steering level not 650 hPa.

Ln154 in the Methods, it is important to fully describe how the composites were formed (i.e., ranges for each cardinal direction). Do the four classes together comprise a subset of the full 2.3M sample set, or a subsample?

Ln173 suggest "for [vertical wind] shear"

Ln 179 re: "accelerate". Relative to what? Shown where?

Ln194 specify source of "continuous lightning observations". This dataset was not included in Lns88-101 study overview

Ln199-223 consider if these paragraphs are just Results or actually Discussion/interpretation of results?

Ln206 suggest "growth is [likely] sensitive" because this is only interpretation

Ln211 suggest "under [vertical] wind shear"

Ln215 re: "strong SM gradients". At what scale length? 100-km?

Ln 222-223 suggest removing "and surface flux patterns" because this was not shown/analyzed

Ln239 remove "controls". It is not shown anywhere that SM controls anything.

Ln257 re: "thunderstorms develop particularly rapidly". Suggest removal. I don't see any analysis of the rate of thunderstorm development. I also think the authors are only looking at CIs.

Ln271 suggest "strong [correspondence of SM-P feedback sign to vertical wind shear]..."

Ln282 re: "SM-P". did all CI events included in the analysis have some minimum threshold of precipitation? This point is not clear.

Ln283 suggest "Assuming a [global] preference"

Ln285 re: "To this list we can add [vertical] wind shear..." To role of wind shear was established by Findell and Eltahir (2003c), if not well before. Regional soil moisture gradients are also widely known to interact with low-level jets and influence the positioning of associated storms (e.g., Campbell et al. 2019).

Ln287 does intended meaning change if the words "expected from" are removed?

Ln294 re: "first time the important mediating effect of wind shear on SM-CI relationships". Suggest the authors have instead contributed to understanding of the effect of SM gradients on wind shear-CI relationships

Ln295 re: "influencing the vertical growth of storm clouds". Where is this result shown? Was this derived or inferred from cloud top cooling rates?

Ln298 re: "limited to drier months of the year". Where is this result shown?

Ln308 re: "North Africa". Is this outside of the study domain (i.e., south of 25N)?

Ln316 suggest "surface [SM] heterogeneity"

Ln318-320. Consider if it is because they are non-linear interactions, or more fundamentally, because models don't represent surface spatio-temporal heterogeneity (incl. phenology) and suffer from inaccurate boundary conditions in Africa where there are sparse observations?

Ln323 suggest "markedly stronger [when paired with a particular configuration of directional wind shear]"

Ln326 "within local dry soil" or "[over] dry soil"?

Ln327 reconsider use and definition of term "explosive"

Ln331 suggest “models could [theoretically improve] early warning”
Ln348 How is parallax corrected? This is the Methods section, afterall.
Ln359 “[on] a semi-empirical”
Ln361 suggest “[incremental] improvements”
Ln375 suggest “[vegetation root] access to groundwater”
Ln387 suggest “[precipitation] gauge”
Ln389 suggest “microwave [radar] are”
Ln391-393 what fraction of total CIs analyzed in Fig 1 are analyzed for lightning?
Ln400 why interpolate for 100-m winds rather than pick nearest neighbor?
Ln402 define angle ranges for each angle class
Ln403 suggest “surface [SM heterogeneity], we”
Ln426 define subscript “sh”. Add definition of “favourable” and “unfavourable” to the Fig. 2 caption. Readers should not have to read the Methods to understand the main figures.
Ln433-436 re: Results from...In summary, the study”. This is confusing that we are now discussing results from another study, or are they also used in this study. If so, specify more clearly where these results from another study are used/shown in this study and how it relates to New results (Figs 1-3) from this study.
Ln443 “Sc” is undefined.
Ln444 “5deg [resolution]”
Ln445 “[CI] events.”
Ln473 re: “reduced sensitivity”. The authors have not calculated sensitivity. Is it correlation or correspondence?
Ln493 should the dataset DOIs be included instead?
Ln504 it should be specified if the available code covers all analyses and figures, or a subset

FIG.1.

Specify time period and source of data for SM, P, CI, elevation, and cloud cooling rate.
Should it be specified these are “daytime-only” CIs
Color bar should have a white increment (i.e., non-blue and non-red) for some +/- offset from 0. As is, a value of +0.00001 or -0.00001 would have red and blue shades, when they are very similar.
Is the color bar range truly [-1%, 1%]?
Ln131 inappropriate use of “anomaly”? anomaly implies difference relative to climatology. should it be “difference”?
Fig. why are the three dashed boxes superimposed on Fig.1a not also included in Fig. 1b?
Ln133 define “low-level”. 100-m.
Ln133 rephrase “upwards”. consider, “to top of page” or similar, so as not to be confused with positive vertical motion.
Ln135 define “local” with a spatial scale length
Fig 1c. why is the x-axis non-linear?
Ln136 re: “topographic standard deviation”. At what scale is this calculated? Is it a 200km radius of the primary CI location, or different?
Ln138 suggest “based on a [precipitation] threshold”
Ln138 re: dSM. Why is the solid box in Fig. 1a not dSM=0?
Ln140 how is this 50-km standoff distance chosen. The choice is not justified in the Methods. Also, what is the dimension of these three boxes over which SM is spatially averaged?

FIG. 2.

Color bar label and units are missing.
Ln156. It does not appear “As in Figure 1”, because the fields have not been rotated as they were in Fig. 1 (i.e., wind flow is not always towards the top of the page). Also, there are not three superimposed dashed boxes.
Ln157,162 “SM” or “dSM”?
Ln158-159 it should be clarified that there is a range of flow directions included in the composite, not just flow in the cardinal directions denoted by the vector plotted in the lower left corner of each panel plot
Ln159 is 3-hour accumulation window leading, lagging, or centered on CI event?
Ln161 suggest “correlation coefficient between SM and P”
Ln162 shear “class” or “configuration”?
Ln163 specify July 2024 to what end date (December 2024)?
Ln163 flashes over 3-hour period centered on CI event, or leading, or lagging?
Ln163 Cloud [top] cooling rate
Ln165 define “favourable” and favourable for what?
Why is the line of symmetry in SM along the diagonal (100km downwind to 100km crosswind) in Fig. 2e?
As an aside, it would be interesting to investigate the diurnal cycle frequency for each low-mid level vertical wind shear configuration.

FIG. 3.

It is not entirely apparent that these simple schematics are fully-supported by the results shown in Fig. 2. Some representative cases drawn from the CI events analyzed may be more effective.
Ln226 suggest “schematic of observationally-supported daytime soil moisture affect on convective initiation location under three vertical wind shear configurations” or similar. The phrase “soil moisture-controlled” should be removed.
In Fig3b legend, is it only “SM-driven circulation” or is it “terrain” or “surface flux gradient- or surface heating-driven circulation”?

FIG.4.

Ln260 is this dSM still defined as the difference between three box-averaged SM at 50-km distance (i.e., Fig.1) It is unclear. Ln268 suggest “shown as black [unfilled or hollow] bars” or similar.

Additional references of relevance to consider citing:

Ford, T. W., Steiner, J., Mason, B., & Quiring, S. M. (2023). Observation-driven characterization of soil moisture-precipitation interactions in the central United States. *Journal of Geophysical Research: Atmospheres*, 128, e2022JD037934.

<https://doi.org/10.1029/2022JD037934>

Zhang, M. S., D. D. Turner, and D. Entekhabi, 2025: Relative Contributions of Surface Parcel and Atmospheric Environment to Convective Potential during Interstorm Periods. *J. Hydrometeor.*, 26, 1395–1406, <https://doi.org/10.1175/JHM-D-25-0045.1>.

Campbell, M. A., C. R. Ferguson, D. A. Burrows, M. Beauharnois, G. Xia, and L. F. Bosart, 2019: Diurnal Effects of Regional Soil Moisture Anomalies on the Great Plains Low-Level Jet. *Mon. Wea. Rev.*, 147, 4611–4631, <https://doi.org/10.1175/MWR-D-19-0135.1>.

Findell, K. L., and E. A. B. Eltahir (2003), Atmospheric controls on soil moisture-boundary layer interactions: Three-dimensional wind effects, *J. Geophys. Res.*, 108, 8385, doi:10.1029/2001JD001515, D8.

(Remarks on code availability)

Version 1:

Reviewer comments:

Referee #3

(Remarks to the Author)

This is a review of the revised manuscript titled, “Wind shear enhances soil moisture influence on rapid thunderstorm growth.” In general, the authors have done a fine job addressing the myriad of concerns raised by reviewers in the first round. Line numbers were omitted from the revision, however, which made writing this review more cumbersome.

I have two final major suggestions: 1) reconsider the title and 2) underscore uncertainty in the ERA5 winds used to compute shear.

With regard to the title, readers will find a disconnect between the title and the core thrust of analysis presented. The title reference to “rapid thunderstorm growth” is really pointing only to a single subpanel (i.e., Fig 2f) and Extended Fig. 6. The heart of the robust analysis is on convective initiation, not cloud cooling rate. Also, the title states that “wind shear enhances soil moisture influence.” The paper shows that wind shear can “modulate” or “mediate”, but not always “enhance”.

With regard to ERA5 winds, the authors wrote in their Response that they were “not aware of any such evaluation [of winds] due to the sparsity of wind observations (even at 10m) across the study area.” Using a dataset of unknown accuracy is a major limitation of this work. Given representativeness of ERA5 low-to-mid level wind shear is central to the authors’ conclusions, they should consider disclosing its accuracy over sub-Saharan Africa is indeed a question mark. One way to do so, is to add a statement to the articles closing paragraph underscoring the need to verify low-level winds upon which the findings depend; improving forecasts of CI location is not only about additional incorporation of land surface state information to forecast systems.

I also have a few minor comments:

1) Sensationalism. My suggestion is to reconsider the unnecessarily sensational term “explosive” in the manuscript. A >55C/hr cooling rate is “explosive”, but not 50C/hr? It’s just a bit over dramatic.

In another place, the authors use a sensational 68% figure, when they could instead explain the difference is ~2%: “the probability of a CI being classified as extreme is 68% higher for favourable”, in Fig. 2g, it looks like a much less remarkable difference of ~2% vs. ~4%.

2) The second sentence of the Discussion could be misleading. As written, the dependence of “SM patterns strongly influence the vertical growth of storm clouds” (Fig 2f) and “favorable SM patterns producing more lightning and rainfall” (Figs 2d,e,g) on coincident favorable wind shear conditions is not strictly clear. Suggest rephrasing.

3) Later in the Discussion, “Favourably aligned [] SM-induced circulations...These are cases for which there is little indication of the imminent storm from satellite imagery even an hour ahead...” As written, this suggests the statement only holds for favourable SM cases, when it likely holds for some unfavourable SM cases, too. Suggest rewording.

4) Authors should double check references to Extended Figures cite the correct Extended Figure number. Also, intended figure references may be missing from the statement, “We use the same approach to create composite maps based on cloud-top temperature (at 5km) and lightning (at 10km).” Above that line, there is a small inaccuracy where it states “We degrade...”, when really ECMWF served the data at the degraded resolution.

(Remarks on code availability)

Response to reviewers

We are very grateful for the time and effort that the reviewers have spent in considering our manuscript. We believe that the revised version is much improved thanks to their input.

Below we provide a point-by-point response to the three reviews. The reviewers' comments are shown on a grey background and line numbers (LXXX) refer to the tracked changes document.

Note that in revising the manuscript and responding to the reviewers' comments we spotted a minor inconsistency between the method described in the text and the calculations we used (L90). Our analysis of initiations (from images up to 1745 UTC) included 114,244 (4.8% of the total) to the east of the Greenwich Meridian corresponding to "evening" initiations (1800 LT or later). In the revision we have repeated all our calculations with a slightly reduced afternoon-only (1200-1800LT) dataset. Our results differ from the original only in fine details (Figure R1). All our figures and statistics in the revised manuscript have been updated accordingly.

Figure R1 Impact of filtering out initiations that occurred from 1800LT onwards. Here we have reproduced Figure 1c (a), 2f (b) and 2g (c) using our filtered initiation dataset (as in the revised manuscript; solid lines) and the unfiltered dataset (as in the original manuscript; dashed lines).

Referee #1 (Remarks to the Author):

This paper identifies a huge number (2.3millions) of deep convective initiation (CI) events over most of Sub-Saharan Africa (with the exception of very wet regions/seasons) and creates composites of Soil Moisture (SM) and rainfall (P) that are stratified by the background directional wind shear.

They show that

i. inhomogeneous SM patterns are associated with the CI events composites for all cases (away from topography) and reach maximum SM dry/wet differences of about 1% (for context most of the CI events, and in particular the extreme ones are occurring in places/seasons when the background SM is of the order of 50%, at least that's how I see Ex.Fig 2)

ii. For optimal SM inhomogeneity, Extreme CI events (whose cloud deepening is characterized by the fastest cooling rates) double (for the highest shear cases) from 3% to 6%, compared to the least favorable SM inhomogeneity (10th to 90th percentile). On average the difference in cooling rate between favorable and unfavorable SM is small for weak shear, but gets to 5-10°C/hr at high shear, equivalent to the effect of doubling CAPE from about 1000 to about 2000J/kg.

iii. The precursor warm cloud and the rainfall associated with the composite CI events are placed at the edge of the dry soil patch for most cases, but at the center of the dry patch when the low level and upper level winds have opposite direction.

iv. (iii) implies that the spatial correlation between rainfall and soil moisture in a 2°x2° box around the CI is positive for most cases, but negative for the strongest directional shear.

v. Across the continent, CI occurs very consistently over dry patches, but the wind has opposite direction between the upper and lower levels in a minority of cases, mostly confined to Tropical North Africa (mean cosine similarity gets to -.5 there, according to Fig 4c); this implies that the negative spatial correlation between SM and P at the time of CI is mostly limited to the latter region (Sahel+Gulf of Guinea).

We would just like to clarify an issue raised in point (iv) about the relative importance of negative SM-P feedbacks across SSA, which the reviewer states is “positive for most cases”. When we split the initiations by directional shear, for the v650+ case (mid-level flow aligned with low level; Fig. 2a), precipitation tends to be favoured over wetter soils, with $r(\text{SM},\text{P})=+0.49$. For both cross-shear cases (Fig. 2b,c), the composite rainfall pattern is maximised where the SM gradient is strongest, with neither positive or negative feedback evident at the scale analysed ($|r(\text{SM},\text{P})|<0.05$), whilst the correlation is very strongly negative for the reverse shear case ($r=-0.85$). Based on this partition into 4 shear configurations, only the v650+ cases yield a positive SM-P correlation, amounting to 34.8% of all cases with SM data. We would therefore say that “the spatial correlation between rainfall and soil moisture in a 2°x2° box around the CI is positive” for only one third of all cases, rather than the majority as stated by the reviewer. To

address this potential misunderstanding, we have now added the proportion of total cases contributing to the positive and negative feedback composites at L147 and L149

“(v650+; 34.8% of all cases)” and “(v650-; 23.5% of cases)”

All of this is done very convincingly, with great care, with the T’s crossed and the I’s dotted. The paper is well written and well illustrated. It is a very solid and very interesting piece of science. What I am not completely sure that I understand are the significance statements made by the authors. Specifically, there are two things I am confused about: how this helps with predictability, and what this means about land/atmosphere feedbacks more broadly.

I see how these results could be a source of predictability for initiation at least in a statistical sense: if one had the ability of processing the SM signals to find the most favorable patterns, one could warn of enhanced probability of extreme CI events, distinguishing between locations 100km apart. What I am not so sure about is whether that would also translate in the ability to forecast beyond the initiation, to the development of the MCS that typically are the systems that are the most impactful. (This is of course leaving aside the issue of how a nowcast of this sort would be not just produced, but also disseminated... that’s for the social scientists!) There is a conclusion here (lines 317-318) that the more rapid initiation favors rainier and more long lived storms, but it’s not self evident to me and there is no citation for the statement.

We agree with the reviewer that these results could provide “a source of predictability for initiation at least in the statistical sense”. We would however like to clarify that the spatial scale of this potential predictability is more like 10-40 km than the 100 km suggested by the reviewer. Strong real-world SM heterogeneity on scales detectable by the MSG LST dataset we use (3km) is common during the rainy season within semi-arid SSA, as can be seen on our near-real time portal (<https://africa-hydrology.ceh.ac.uk/nowcasting/>). From previous (“shear-blind”) analyses we know that SM contrasts on scales of 10-40km provide the dominant signal on CI, and we now refer to this length scale range in the introduction (L66)

“spatial variations in surface fluxes on scales ~ 10-40 km”

The reviewer makes a good point with respect to the prediction of subsequent MCS development. In previous work we have looked at aspects of how the land surface influences MCS intensity, notably (in the Sahel) Klein and Taylor (PNAS, 2020) and (across global MCS hotspots) Barton et al (Nature Geosci, 2025). Once a system organises, intense rain can affect a very large area. However, local cumulo-nimbus storms also cause damage, particularly in areas vulnerable to flash flooding. In this study we are focused on a different (and arguably more challenging) aspect of the prediction problem - given favourable larger-scale conditions, where is deep convection most likely to initiate? For this prediction problem we argue that the land surface provides information on key scales of 10-40 km. The question of how that knowledge can translate into improved prediction of mature MCSs is left for other studies.

We have modified our final paragraph to emphasise that our work targets predictability of storm initiation, which is a key gap in developing early warnings of storm hazards on user-relevant scales (L341-345)...

“Our results indicate that incorporation of land surface state information into AI-based or Numerical Weather Prediction models has the potential to improve fine-scale prediction of thunderstorm initiation. This addresses a major limitation in the provision of early warnings of hazardous storms ...”

The reviewer also asks about the basis of a statement on (original) lines 317-318 in a paragraph discussing the highly-sheared environment across Tropical North Africa. We said “the combination of wind shear and surface heterogeneity [in Tropical North Africa] generates particularly rapid cloud growth in the early stages of storm life cycle, favouring rainier and likely also longer-lived thunderstorms”. We have evidence from the 3-hour rainfall accumulations in Figure 2, and also composite maps showing the likelihood of heavy rainfall (not included in the manuscript), that the v650- composite does indeed produce heavier rain (according to the IMERG dataset) than the other three shear configurations. For the sake of brevity however, in response to the reviewer’s comment, we have removed the final part of the sentence (starting “favouring...”) (L327-329).

I am also wondering whether predictability can be determined by compositing on the CI. Once the CI is identified, one can see that it had a warm cloud precursor right at the downwind edge of the dry patch, where the gradients are strongest (or at the core of the dry patch, for the v850-case). But don’t we need to know the complementary: given a dry patch, how likely it is that a CI will occur?

The reviewer raises the point that to determine predictability, one needs to examine the statistics of CIs occurring given a dry patch. In fact, our Extended Data Figure 4 provides some information on this point. It compares the climatological cumulative distribution frequency of ΔSM (sampled in the same locations and on the same days of year as the initiations, but in different years) with the cumulative distribution frequency of ΔSM on initiation days. For example, it tells us that 27.9% of CIs (and 32.8% of extreme CIs) occur for values of ΔSM expected 25% of the time. It thus tells us how much more likely a CI is for a given dry patch, based on the simple ΔSM metric. We have added a sentence in the Methods to address this point (L437-438):

“The curves thus quantify how the probability of CI changes across the climatological distribution of ΔSM .”

It might be that once the shear and the CAPE are there, we already have a good handle on whether CI will occur, and the land information can add some detail... but is this the case? Or is the problem with the forecast skill such that we don’t know IF CI will occur, let alone where exactly?

We believe this is the case, and from experience land information at finer scales does help. We have been working with African Met Services on the nowcasting problem in the last 3-4 years (eg Taylor et al ERL 2022, Fletcher et al, BAMS 2022). They know from NWP and observations when large-scale conditions are favourable. In the Sahelian rainy season (July-Sept) for example, this is closely tied to the phase of African Easterly Waves. On scales ~500-1000 km afternoon storms will develop every day within favoured regions ~500-1000km, zones often associated with AEWs. Taylor et al (2022) discuss the value of land surface information for real time

nowcasting based on experiences of forecasters and researchers working side-by-side in a “Testbed”. For example, we found that a combination of LST/SM heterogeneity and extended convergence lines (identifiable by shallow cloud in satellite imagery) to be valuable pre-cursors. In that paper we wrote “By combining land and cloud observations, such favoured locations could be identified up to 2 h ahead of triggering, giving forecasters a head-start in providing local-scale nowcasts.”

The other issue I am confused about is how we go from the spatial correlation between SM/P for a box around the CI event, to the characterization of the SM/P feedbacks more broadly. My understanding is that the negative spatial correlation might lead to homogenization of the SM field and less likely CI (and vice-versa: a the positive SM/P correlation would be sharpening SM gradients and lead to more likely CI). But how does this translate to temporal SM/P feedbacks, when a lot of the rain comes from remotely triggered systems? On a related note, the connection to the negative autocorrelation of rainfall (at day 1 lag) also seems tenuous, given that the latter seems to be explained by atmospheric waves, and not land processes.

When I say I am confused, it’s not a white lie. I genuinely wonder whether I am missing something here that somebody with a deeper knowledge of the literature would be able to see and that would make them come to the same conclusions as the authors. But at the very least these issues should be clarified and discussed less obliquely.

It is evident from the comments of both Reviewer 1 and 2 that the original manuscript was unclear in making the link between negative spatial and temporal SM-P feedbacks in Tropical North Africa. We have expanded the text to address this point (L319-324):

“Within this region, observed rainfall in the two days after an MCS is favoured over drier areas unaffected by the initial storm (Taylor et al, 2024), implying a negative temporal (as well as spatial) SM-P feedback there. We suggest that shear-mediated land-atmosphere interactions in Tropical North Africa contribute to the globally unusual negative temporal SM-P correlation observed there (Guillod et al, 2015)”

On the specific points Reviewer 1 makes:

“My understanding is that the negative spatial correlation might lead to homogenization of the SM field”

Our 2024 paper quantifies the multiday evolution of the SM field in the aftermath of an MCS. Heterogeneity decreases in the days after a storm as the soil dries. At the same time, new rain occurs, falling preferentially on drier areas. However, this negative spatial SM-P feedback is not so efficient that it wets up all the places that were initially dry, and synoptic processes still influence whether rain can occur at all. The combination of these processes produces a complex evolution of heterogeneity, so we cannot say that the negative SM-P feedback homogenises the surface.

“But how does this translate to temporal SM/P feedbacks, when a lot of the rain comes from remotely triggered systems?”

The reviewer is correct that the majority of rain in the region comes from remotely-triggered MCSs. We find negative SM-P feedback processes operate on mature as well as initiating MCSs, as documented by Klein and Taylor (2020).

“the connection to the negative autocorrelation of rainfall (at day 1 lag) also seems tenuous, given that the latter seems to be explained by atmospheric waves, and not land processes”

We agree that atmospheric waves play an important role in determining the timing of rainfall over West Africa, favouring a negative autocorrelation signal. However, as we noted in our 2024 paper “marked local rainfall suppression over wet soils for 2 days after an MCS can only be explained by the persistent spatial forcing that SM provides”. This spatial SM-P feedback contributes to the negative autocorrelation.

Other minor comments:

L 117: Extended figure out of order.

We have reordered several of the Extended Data Figures so that they are now numbered in order of being referred to in the text.

L 119: a couple of words on how you define the shear and a reference to Methods would be nice

We now include some words on our definition of shear when we introduce the data (L97-98)

“with shear defined as the vector difference between winds at these levels”

L209: I am a bit unsure of what you mean here by favourable. Are you making a general statement, or are you referring to your favourable/unfavourable SM patterns? Are you suggesting that the updraft is moving from t0-1 hour to t0 to a more favourable environment?

We mean “favourable” in a general sense, as expanded on in the subsequent text “where wider, more entrainment-resistant updraughts can develop”

L221: how are your result consistent with the sensitivity to updraft width? What can you say about updraft width?

While we do not directly observe updraught width, we show more rapid vertical growth under higher shear, further enhanced by favourable SM gradients (Fig.2). Prior studies link updraught width, vertical cloud extent, and cloud relative inflow via mass continuity (Peters et al, 2022a,b), and show that wider updraughts are more resistant to shear-enhanced entrainment (Marquis et al. 2021, Morrison et al. 2022). We cannot explain the observed deeper, faster-growing CIs under stronger wind shear without inferring increased updraught widths. Likewise, we assume that favourable SM gradients support wider updraughts via enhanced cloud relative inflow, explaining the observed sensitivity in vertical cloud development.

In the revised text we have restructured the sentence flagged by the reviewer (L223-228):

“This explanation for our satellite observations is consistent with existing observational⁴¹ and model^{42,43} studies that note enhanced deep convective development associated with terrain

features, mesoscale circulations and surface flux patterns, where resulting larger initial updraught widths can better withstand increased entrainment under sheared conditions¹⁴.”

L245: I would add a couple of words (and a reference to Methods) to indicate that you composite over all CI events occurring in 2°x2° boxes.

We have added to this sentence (L251):

“between SM and P composites created every 2°”

L255-256: for clarity, I would say “negative SM-P spatial correlations during CI events, particularly...”

We have edited L261:

“negative spatial SM-P correlations”

L284: Can you add a reference for this? It makes sense, but I did not know where to read about the effect of typical lifetime convection

We were not referring to a particular study, rather the logic that a longer-lived system will travel further from its initiation point and may leave the locally drier area. We replaced “it is known to depend on” with “it will depend on” (L290).

L286: globally, rain occurs more often than expected from climatology for the most negative cosine similarity, but the same figure shows the opposite for SSA. Am I missing something?

The statement refers to the global analysis. The reviewer raises an interesting point with respect to SSA. The first bin in Figure 4h, representing the frequency of strongly negative values of cosine similarity on event and non-event days does indeed show distinct behaviour for SSA (magenta) compared to the 7 remaining bins. This particular sample is dominated by Tropical North Africa during Northern summer. We have not investigated this but suggest it could be due to a negative correlation between v650- and unfavourable thermodynamic conditions. To address the comment we have added a “tend to” (L292)

“afternoon rain days globally tend to occur on days with stronger directional shear”

L317-318: Why rainier and more long-lived?

We have now removed this phrase.

Referee #2 (Remarks to the Author):

The manuscript “Wind shear enhances soil moisture control on rapid thunderstorm growth” presents a novel, data-rich, and physically consistent analysis of how wind shear modulates the impact of soil moisture heterogeneity on convective initiation (CI) across sub-Saharan Africa. Using an unprecedented dataset (~2.3 million CI events) and integrating high-resolution satellite observations with ERA5 reanalysis and IMERG precipitation products, the authors convincingly show that the most explosive storm initiations preferentially occur over locally dry soils when wind shear is strong. They further demonstrate that the direction of shear determines the sign and magnitude of soil moisture–precipitation (SM–P) feedbacks.

This study represents a major advance in understanding where and why thunderstorms develop under similar large-scale conditions, providing a mechanistic explanation for negative SM–P feedbacks observed in the tropics. Its implications extend globally, with clear relevance for convective predictability and the evaluation of convection-permitting climate models. The analysis is methodologically rigorous, grounded in theory and observations, and clearly written. Overall, this is an excellent and highly original contribution. It convincingly demonstrates how mesoscale soil moisture contrasts interact with vertical wind shear to modulate deep convection and offers a robust, physically grounded explanation for the observed spatial SM–P relationships. I consider this paper suitable for publication in Nature pending minor revisions to improve clarity and discussion.

Major Comments

1. The definition of “favourable” and “unfavourable” SM configurations (dSM_{sh}) could be expanded. While the manuscript references Extended Data Fig. 6, it would help to explicitly define how thresholds were chosen.

We now explicitly state our motivation for choosing a length scale of 50 km for computing ΔSM (L430-431).

“We chose a distance of 50 km based on previous analysis highlighting SM gradients on length scales of 10-40 km”

We have modified the text to clarify how we chose the box locations for computing ΔSM_{sh} (L44--441)

“These locations (plotted in Extended Data Fig. 5) are selected to capture the dominant gradients in the composite-mean SM patterns shown in Figure 2a-d”

We chose to create the samples based on the first and tenth deciles to ensure sample sizes were large enough to produce robust results.

2. It would be valuable to provide an uncertainty estimate or sensitivity analysis for SM retrieval errors and their impact on dSM values, especially given known retrieval challenges over dry, sparsely vegetated regions.

In response to this comment we now include a map of the mean uncertainty in the SM signal (Extended Data Figure 1g). We have added the following in the Methods section (L379-380):

“Over the majority of SSA, uncertainty in the retrieval of SSM is less than 4% (Extended Data Figure 1g).”

In Figure 1c we already include error bars denoting the standard error of the mean value of dSM (now Δ SM) per decile of topographic variability. This measure of uncertainty implicitly includes errors from the SM retrieval, sampled many tens of thousands of times. It is clear from this figure that the signals in Δ SM lie well above the noise.

3. Clarify how parallax corrections were handled quantitatively (e.g., typical offset distances).

We have added the following text (L360-362)

“...using climatological profiles from ERA5 to convert cloud-top temperature to height. This results in eighty eight percent of cases with offset distances of 10 km or less.”

4. The discussion would benefit from a more explicit contrast between spatial and temporal SM-P feedbacks. The manuscript notes that spatial feedbacks are strongly negative under shear, but the implications for daily or subseasonal memory of rainfall could be elaborated.

We have added a sentence linking spatial and temporal SM-P feedbacks with Tropical North Africa (L319-321). Implications of the spatial feedback on subseasonal memory of rainfall is specifically addressed by Taylor et al (2024).

“Within this region, observed rainfall in the two days after a storm is favoured over relatively dry areas (Taylor et al 2024), implying a negative temporal (as well as spatial) SM-P feedback.”

5. The final discussion paragraph mentions implications for nowcasting and AI-based systems. This is an excellent point. Consider expanding with one or two sentences on how these findings might be operationalized (e.g., coupling near-real-time SM retrievals with convection-permitting nowcasts).

We have expanded the sentence to read (L341-343)

“Our results indicate that incorporation of land surface state information into AI-based or Numerical Weather Prediction models has the potential to improve fine-scale prediction of thunderstorm initiation.”

6. The study could briefly acknowledge the absence of direct in-situ flux or sounding data to confirm the proposed mechanisms (entrainment resistance, convergence strength).

We now specifically mention that our interpretation of the data is based on satellite observations (L223).

7. It would be helpful to note that ASCAT SM measures only the top few centimeters of soil—thus the coupling may differ under conditions of strong vegetation control or sub-surface moisture gradients.

We already mention when introducing the data that we are using *surface* soil moisture measurements (L93), whilst the specific depth is mentioned in the Methods (L374). It is true that where surface fluxes are more influenced by strong vegetation or sub-surface moisture controls we would not be able to see those effects. However, in our previous study over West Africa (Taylor et al 2024) we found that the memory of an MCS on subsequent rainfall patterns

could only be detected for two days afterwards, and out to 8 days under drier conditions. On those time scales a memory of the MCS was evident in the ASCAT surface SM dataset.

Minor Comments

1. In the Introduction, when mentioning afternoon storm development, cite relevant diurnal cycle studies.

We are limited by the journal guidelines to 50 references.

2. In the Results, clarify that “extreme” refers to rapidly cooling cloud tops rather than the most intense storms by rainfall or lightning.

The second paragraph of the Composite Analysis section begins with our definition of extreme events (L115-116)

“We define extreme CI events as those in the top 1% of cloud cooling rates, corresponding to -78 °C hr^{-1} or more”

3. In Figure 3, define timing conventions (e.g., t_0 , $t_0-1\text{h}$) within the caption for readers less familiar with CI detection.

We have adjusted the text (L230)

“Shallow convection an hour before initiation ($t_0-1\text{h}$)”

4. Ensure consistent use of abbreviations (e.g., “CI” vs “CIs”).

We have gone through the manuscript in a consistent manner and identified where we should refer to the singular (CI) or plural (CIs).

Recommendation

Minor revision: This is an exceptional and timely paper offering a significant conceptual advance in land–atmosphere coupling and convective dynamics. The suggested clarifications would further strengthen its clarity and accessibility to a broad readership.

Referee #3 (Remarks to the Author):

This is a review of the manuscript titled, “Wind shear enhances soil moisture control on rapid thunderstorm growth”, submitted to Nature by Taylor et al. The manuscript details the authors’ analysis of vertical (100m- 650 hPa) directional wind shear and convective storm initiation relationships conditioned on mesoscale surface soil moisture gradients. The analysis focuses on sub-Saharan Africa over a 21-year record using: Meteosat Second Generation land surface temperature, cloud top temperature, lightning data; Scatterometer surface soil moisture estimates; NASA IMERG satellite-based precipitation, and ERA5 100m and 650 hPa reanalysis winds. The authors find that the vertical directional wind shear can partly explain the difference in sign and magnitude of previously-reported soil moisture-precipitation correlations that did not consider vertical directional wind shear. In particular, focusing on the most extreme 1% of storms, “...the greatest vertical storm growth occurring where soil moisture-driven circulations oppose the direction of shear-induced cloud displacement. Developing clouds follow the mid-level wind direction, and where this opposes the low-level flow, rainfall is strongly correlated with locally drier soils.” These findings are significant and novel enough to warrant reporting in Nature, and moreover, the analysis is an exemplary application of multiple geostationary and low-earth orbit datasets. However, there are many minor, but important issues that should be addressed prior to publication. These are outlined below in order of line number.

Within the constraints of the journal format, we have responded as best we can to the reviewer’s many helpful comments. Note that this includes a limit of 50 references in the main text.

Overall, 1) all figure captions need to be revised for clarity,

We detail our numerous changes to figure captions in the point-by-point response below.

2) Figure 3b’s schematic should be improved,

We discuss this comment in the point-by-point responses.

3) the added-value of Fig. 4d-h and related analyses to the discussion/findings should be re-evaluated. If Fig. 4d-h are truly important and central to the authors' message, then the text, discussion, and figures should be reworked to elevate these findings accordingly.

The reviewer considers the added-value of Figure 4d-h. Those panels follow on from Figure 4(b,c,f), which demonstrate how directional wind shear plays a major role in determining the sign and strength of SM-P feedbacks across Sub-Saharan Africa (SSA). This is one (of several) important results in our regional analysis. In Fig 4 (d,e,g and h), we have placed this particular finding in a global context by re-examining results from a previous study through the lens of wind shear. We find that “whilst directional shear outside SSA tends to be much weaker, shear still plays a mediating role in determining the [global distribution of] spatial SM-P feedback” (L287-288). We consider that this result is “truly important” as it (a) gives global context to our African analysis, (b) reinterprets results from a previous, highly-cited paper in Nature, and (c) provides an explanation for why SM-P feedbacks in Tropical North Africa, both spatial (Taylor et al 2011, Guillod et al 2015) and temporal (Taylor et al 2024, Guillod et al 2015), stand out from other regions.

The reviewer suggests that if this result is retained, then “the text, discussion and figures should be reworked”. The principal theme of our work is on the important role of wind shear in understanding how SM influences where thunderstorms initiate (or alternatively, how SM influences shear-CI relationships). This focus on CI (rather than SM-P feedbacks) is reflected in the title and the majority of the text. The implications of this behaviour for the geographical distribution of SM-P feedbacks (across both SSA and globally) is the subject of one of our four figures (Figure 4), based on analysis of SM and P data split by wind shear in Fig 2(a-d). We devote a paragraph in the Geographical Variability section to the global analysis (L277-296), and include a sentence in the Summary paragraph based directly on the material that the reviewer highlights – “Whilst such shear conditions are particularly common over Tropical North Africa, the effect favours negative soil moisture-precipitation feedbacks globally.” (L21-23). However, we recognise that the original discussion section only referred to the global analysis implicitly. In response to the reviewer’s comment, we have now reworked the text in the discussion section from a more explicitly global perspective (L318-324).

“The shear-based mechanism provides a dynamical explanation for why the region exhibits the strongest negative spatial SM-P feedbacks globally. Within this region, observed rainfall in the two days after an MCS is favoured over drier areas unaffected by the initial storm, implying a negative temporal (as well as spatial) SM-P feedback there. We suggest that shear-mediated land-atmosphere interactions in Tropical North Africa contribute to the globally unusual negative temporal SM-P correlation observed there, with widespread negative lag-1 daily precipitation autocorrelation.

Responses to line-by-line comments

Ln1/Title: The title should be revised to better represent the actual scope of work. For example, “Vertical wind shear enhances soil moisture affects on daytime convective initiation in sub-Saharan Africa” or “Vertical wind shear affects the role of regional soil moisture gradients on daytime convective initiation in sub-Saharan Africa”, or similar. The terms “control” and “rapid thunderstorm growth” are unsupported by the analysis.

The reviewer makes some good suggestions for better aligning the title to the scope of the work. However, there is a 75-character limit on titles in Nature which means that we cannot consider refinements such as “sub-Saharan Africa” or indeed “vertical (wind shear)” without losing the three key elements of wind shear, soil moisture gradients and thunderstorm growth. In the revision, we have replaced the word “control” with “influence” in the title. We prefer to retain “rapid thunderstorm growth” however. Whilst we agree with the reviewer that not all CIs result in thunderstorms (comment on L26 below), on average the CIs that we analyse produce considerable lightning activity (Figures 2e and Extended Figure 7), with more rapid growth (e.g. the extreme events in Extended Figure 5j) expected to produce more lightning.

the Abstract should clarify the type of wind shear (i.e., wind speed, direction, or both, with height)

The original summary mentions that we are considering wind differences with height (“whilst wind shear between low and mid-levels”, L14). The original summary also refers to directional effects (“with greatest vertical storm growth occurring where soil moisture-driven circulations oppose the direction of shear-induced cloud displacement”, L18). Though we tried, we are unable to additionally capture the relationship with shear magnitude (as shown in Figure 2f)

within the confines of a 200-word limit of the Summary paragraph without losing other key elements.

Ln17 re: “sensitive...via an interaction”. The terms “sensitive” and “interaction” suggests a modeling analysis was conducted to perform attribution, which was not.

We have replaced “sensitive to” with “favoured over” (L15). We choose to retain “via an interaction with wind shear” as this interaction is central to our study. We note that all 3 reviewers consider this central message to be robust, based on our sampling of the observational dataset by wind shear direction and magnitude (Figure 2), even in the absence of a modelling study.

Ln25 re: “source of predictability”. Predictability is defined by modeling, not through observation. This analysis did not show “soil moisture heterogeneity and wind shear provides an important source of predictability...”

We now refer to the potential for predictability (L23-24) - “provides a potentially important source of predictability”

Ln26 how well does ERA5 represent thunderstorms? Are all convective initiation events assumed to be thunderstorms? In reality, not all CIs result in thunderstorms.

The aim of our study is not to evaluate the representation of thunderstorms in ERA5. Realistically capturing storms over Tropical Africa is a particularly challenging problem for forecasting systems, as evaluated in the paper by Vogel et al (2020). As noted above, on average the CIs we analyse exhibit strong lightning activity.

Ln 28 suggest “..storms and [their associated] hazards...”

Replaced (27-28)

Ln31 re: “increasing exposure”, exposure to what?

We have added the word “storm” to this sentence to clarify (L29).

Ln35-36 suggest truncation: “Deep convective initiation (henceforth CI) can occur on time scales of tens of minutes (REFERENCE). Updrafts within the cloud...” Here and throughout, the term “updraft” should be used instead of “updraught”.

We have used the Oxford English Dictionary spelling of “updraught” and now include a reference (L35).

Ln38 suggest “particularly in [impervious cover-dominated] urban areas with [rapid, higher-volume runoff (REFERENCE)]

In a longer format journal, we would follow this suggestion.

Ln40 “low-level” not “near-surface”; near-surface is 2-10m AGL; CAPE is not near-surface. Often, MUCAPE is used, which is lowest 300 hPa of the atmosphere.

Changed (L39).

Ln47 “[PBL-top] entrainment”

Here we are not specifically referring to a particular height at which entrainment occurs, so prefer not to add “PBL-top”

Ln57 reword “fixed triggers”. Perhaps, orographic or terrain?

Replaced with “terrain features” (L55).

Ln58 suggest “vegetation [water content] and soil moisture”

Limiting vegetation to water content is somewhat restrictive – excluding for example variability in leaf area index and land cover.

Ln59 suggest “Whilst [land cover classes] can be accurately...” Is it true that cropland and irrigation can be accurately mapped in present-day forecasting systems?

It is true that land cover maps (including classifications for cropland, and in some cases, irrigated areas) are used within current forecasting systems. However, the quality of those classifications, and their translation into model-specific parameters, is open for debate. Our point was to distinguish fixed landscape features from dynamic elements (notably soil moisture). We have rewritten the sentence to focus on properties that can be accurately mapped in space (notwithstanding the above; L57-58).

“Whilst fixed surface properties such as cropland, forest and urban areas can be accurately mapped and incorporated into forecasting systems...”

Ln62 suggest “semi-arid [regions].”

Changed (L60).

Ln 64 suggest “how [the changing] balance” or just “how evaporative fraction”

We’d prefer not to introduce “evaporative fraction” at this point.

Ln76,81 was it intentional to cite ref no. 29 in both cases?

Reference 29 is Guillod et al, which quantifies both spatial and temporal coupling between soil moisture and precipitation. It is entirely appropriate to reference in both places, so yes it was intentional.

Ln82 glaring omission of Koster (2004, Science) reference for “hotspots”

We originally chose not to quote that GLACE study here as we were referring to specific regional observational studies in “hotspot” regions (the Sahel, Europe, the Tibetan Plateau, Argentina and the US Great Plains). The GLACE paper identified three (partially overlapping) hotspots (West Africa, Great Plains and India) based on northern summer global model simulations. However, in hindsight, we agree with the reviewer and now cite it (L79).

Ln86 wind shear-CI relationship is classical topic of exploration. So, what is relatively novel is the addition of satellite-based soil moisture heterogeneity. Important to rephrase accordingly.

We agree that wind shear-CI relationships are a classical topic of exploration. However, the novelty comes in the first part of the sentence (L82-83).

“Importantly, the effect of wind shear on SM-CI relationships has not been explored...”

Ln86-87 for truly “severe” storms, driven by synoptic circulations, it is unusual to claim that soil moisture is the “control”. Most SM-P correlation occurs in warm/dry season with relatively less intense storms.

We never intended to give the impression that soil moisture is the “control” and have removed any suggestion of that in the title (in agreement with the reviewer). In fact, the sentence here (L86/87) emphasises wind shear rather than soil moisture. The point made by the reviewer about the time of year when SM-P correlations occur may be more relevant for mid-latitudes than sub-Saharan Africa. In fact, we show strong evidence of a seasonal coincidence of intense storms and strong soil moisture effects on CI in the bottom two rows of Extended Data Figure 2.

Ln88 the precise study period (i.e., 2007-2024 or 2004-2024) should be specified in parentheses. Relatedly, the available MSG data only appears to support an 18-year analysis (see Ln 367).

The exact years of MSG data are provided later in the paragraph (L89). We had omitted the years of availability for the ASCAT data (now rectified at L94), which as the reviewer points out, limits the period of analysis for soil moisture observations to 18. However, our additional LST analysis is based on 21 years, as indeed are our statistics on extreme CIs.

Ln88 reconsider use of term “imagery”

As we are identifying cloud-top temperature features within a satellite image we are quite comfortable with the term “imagery”

Ln94 suggest “primary” will need to be defined in this context

The original text in the Methods section (L354-357) provides an explanation for what is meant by “primary”. To be in keeping with the style of the journal, we have tried to limit details in the main text as much as possible.

“Initiation events are identified when cold cloud appears for the first time (t_0) within 30 km, with minimum temperatures within that radius having cooled at least $10^{\circ}\text{C hour}^{-1}$ over 4 consecutive time steps. This approach excludes secondary initiations caused by prior convection within 30 km.”

Ln96 suggest “spatial [SM] heterogeneity”

Changed (L92).

Ln100 suggest “[resampled] to 1deg resolution”

We think “degraded” provides additional meaning (going to lower resolution) than “resampled” and have retained the original wording

Ln100 reference to study that verifies/evaluates 100-m ERA5 winds in sub-Saharan Africa would be appropriate here

We agree that it would be good to include a reference here, but we are not aware of any such evaluation due to the sparsity of wind observations (even at 10m) across the study area.

Ln103 suggest “composite of [pre-CI] SM where”

We have changed to “pre-CI” at L101 and L269

Ln104 rephrase. To “align with the low-level wind” of what? Seems to me that both SM and wind fields are rotated so that the prevailing wind becomes southerly or towards the top of the page in Fig.1.

We have rewritten this sentence and hope that it is now clearer (L101-103):

“... we performed a spatial composite of pre-CI SM, having first rotated the SM field around the initiation location so that all cases shared a common low-level wind direction”

Ln106 re: “consistent with those previous studies”. What is important to note is whether the findings are consistent with studies over the shared study domain (i.e., sub-Saharan Africa and Sahel) and include those references.

In fact, we identified a consistent elliptical structure across all the regions (Sahel, Europe, Tibet, Argentina) in those previous studies.

Ln107 the analysis does not “explore sensitivity to controlling parameters of the local SM gradient” because it is not a modeling study

We have rewritten this (L105-107)

“Our vast dataset allows us to better explore how the local SM contrast around CIs varies under different conditions.”

Ln109 suggest “most topographically-complex locations”

Changed (L109)

Ln 110 in the era of AI/ML and hybrid physical models, it is important to clarify the type of idealized modeling. Are these regional climate modeling?

New text (L110) reads “Idealised numerical modelling”

Ln suggest “of [sub-Saharan] Africa” consistent with study domain

Changed (L111)

Ln115 suggest “events as[those in the top 1%] of cloud cooling rates”

Changed (l115)

Ln119 suggest “strong [vertical] wind shear”

Added wording earlier on to define “shear” (L97-98)

“with shear defined as the vector difference between winds at these levels”

Ln123 suggest “[tropical] rain belt” or more descriptive reference

Added (L123).

Ln125 re: amplitude. Rephrase. Fig 1b does not show a wave.

“amplitude” replaced with “magnitude” (L125).

Ln148,149,187, 255, 401 here and elsewhere, suggest replacing “steering” with “mid-level” for consistency and correctness. 500 hPa is considered steering level not 650 hPa.

Changed throughout

Ln154 in the Methods, it is important to fully describe how the composites were formed (i.e., ranges for each cardinal direction). Do the four classes together comprise a subset of the full 2.3M sample set, or a subsample?

We have rewritten the text in the methods (L416-418) to clarify that each event is assigned to one of 4 directional classes.

“We assign each case into 1 of 4 directional wind shear classes, based on the angle of the mid-level wind relative to the low-level.”

Combining the 4 classes in Figure 2a-d is equivalent to the all-case composite in Figure 1a. Note that the numbers of events in each composite are provided within the relevant panel. There are only 719,120 cases (out of a maximum of 2.2M) in Figure 1a because many events have no SM data on the morning of the event, are spatially incomplete in the vicinity of the initiation, or occur in 2004-2006, before ASCAT data are available. To the original wind vector plotted for each shear configuration in Figure 2, Extended Data Figures 3 and 7, we have now added a shaded quadrant to provide a visual reminder of the range of mid-level flow angles involved.

Ln173 suggest “for [vertical wind] shear”

Added wording earlier on to define “shear” (L97-98).

Ln 179 re: “accelerate”. Relative to what? Shown where?

We have added text to clarify this sentence (L180-181):

“accelerate vertical cloud growth compared to the all-case mean (Figure 2f),”

Ln194 specify source of “continuous lightning observations”. This dataset was not included in lns88-101 study overview

We have addressed this by introducing the lightning data in the study overview and shortening later text to minimise repetition:

“whilst continuous lightning data from Meteosat Third Generation (MTG) is available from July 2024” (L95-96)

“The 6 months of available continuous lightning observations...” (L196-197)

“We analyse level 2 Lightning Flash data from MTG for the final 6 months of our study period, with data put on a regular 0.05° grid.” (L404-408)

Ln199-223 consider if these paragraphs are just Results or actually Discussion/interpretation of results?

This text is now in a sub-section headed “Mechanistic Interpretation” (L202)

Ln206 suggest “growth is [likely] sensitive” because this is only interpretation

Added “likely” (L210)

Ln211 suggest “under [vertical] wind shear”

Added wording earlier on to define “shear” (L97-98).

Ln215 re: “strong SM gradients”. At what scale length? 100-km?

We have now added reference to length scale analysis from earlier studies, suggesting the strongest effects on CI occur for SM scales ~10-40 km

“primary role for spatial variations in surface fluxes on scales ~ 10-40 km” (L66)

Ln 222-223 suggest removing “and surface flux patterns” because this was not shown/analyzed

The study quoted here (by LeBel and Markowski) did in fact impose a strip of enhanced sensible heat flux, so we retain the reference to surface flux patterns.

Ln239 remove “controls”. It is not shown anywhere that SM controls anything.

Replaced “controls” with “influences” (L244)

Ln257 re: “thunderstorms develop particularly rapidly”. Suggest removal. I don’t see any analysis of the rate of thunderstorm development. I also think the authors are only looking at CIs.

Extended Data Figure 2p-s shows the locations of the extreme CIs, defined by the fastest vertical cloud growth, with a concentration in Tropical North Africa. Extended Data Figure 5j shows that 1 hour after CI, minimum cloud-top temperatures ~-80°C are the norm within this subset of cases. Given the very marked local lightning activity in Figure 2e and Extended Data Figure 6), we believe that the thunderstorm description is entirely appropriate here.

Ln271 suggest “strong [correspondence of SM-P feedback sign to vertical wind shear]...”

Changed (L277-278)

“a strong relationship between SM-P feedbacks and shear”

Ln282 re: “SM-P”. did all CI events included in the analysis have some minimum threshold of precipitation? This point is not clear.

We are uncertain if the reviewer is referring to the current study (where no precipitation threshold is applied) or the global afternoon rain analysis (where a threshold of 3 mm was used). We have now added that latter detail in the Methods section

“in rain events (minimum 3 mm) that develop during the afternoon” (L452-453)

Ln283 suggest “Assuming a [global] preference”

Changed (L289)

Ln285 re: “To this list we can add [vertical] wind shear...” To role of wind shear was established by Findell and Eltahir (2003c), if not well before. Regional soil moisture gradients are also widely known to interact with low-level jets and influence the positioning of associated storms (e.g., Campbell et al. 2019).

We thank the reviewer for reminding us of the study by Findell and Eltahir, which examined the

sensitivity to wind conditions of their 1D thermodynamic framework to the soil moisture-convection issue. That analysis, based on numerical modelling, and assuming homogeneous land surface conditions, found that wind shear could affect the likelihood of convection (consistent with many other studies), but “when the winds allowed, convection occurred in a manner consistent with the 1D-based expectations.” Our observational analysis considers soil moisture (strictly, soil moisture anomalies from climatology) in the vicinity of CI, relative to the mean soil moisture conditions over a 400 x 400 km domain. The fact that here, and in our previous regional analyses, the dominant spatial structures we find are dry soil areas on scales of the order of several tens of kilometres, leads us to conclude that heterogeneity on this scale is more important than broader (sub-200 km) uniform soil moisture in determining the likelihood, and especially location, of CI. This runs counter to a 1D (implicitly homogeneous) thermodynamic approach, even when considering wind conditions, as in the Findell and Eltahir paper cited by the reviewer.

We now add a citation to this specific paper (L309-310)

“the underlying mechanism in this case bares no relationship to widely-used thermodynamic arguments (Findell and Eltahir, 2003a), even when explicitly considering wind shear (Findell and Eltahir, 2003b).

On the topic of regional soil moisture gradients influencing storms via low-level jets and wind shear more generally, we did not cite that literature in this instance as we are focused on convective initiation at small spatial scales given larger scale wind conditions.

Ln287 does intended meaning change if the words “expected from” are removed?

Changed (L293)

Ln294 re: “first time the important mediating effect of wind shear on SM-CI relationships”. Suggest the authors have instead contributed to understanding of the effect of SM gradients on wind shear-CI relationships

This is a very interesting comment. We agree with the reviewer that we have contributed to understanding of SM gradients on wind shear-CI relationships, but at the same time we have also advanced understanding of how soil moisture influences CI.

Ln295 re: “influencing the vertical growth of storm clouds”. Where is this result shown? Was this derived or inferred from cloud top cooling rates?

This result is shown in Figure 2f, based on cloud-top cooling rates for all, favourable, and unfavourable soil moisture configurations. Indeed Reviewer 1 notes that the difference between favourable and unfavourable conditions (taken from the first and tenth deciles of the distribution) is “equivalent to the effect of doubling CAPE from about 1000 to about 2000J/kg”.

Ln298 re: “limited to drier months of the year”. Where is this result shown?

We now refer explicitly to Extended Data Figure 2 (L304) where this result is shown.

Ln308 re: “North Africa”. Is this outside of the study domain (i.e., south of 25N)?

The heat low straddles the northern boundary of our domain during July-September but is located further south at other times of year (Extended Data Figure 2). The convection that we are

referring to in Tropical North Africa is to the south of the heat low, where strong wind shear and CAPE are combined.

Ln316 suggest “surface [SM] heterogeneity”

Changed (L327)

Ln318-320. Consider if it is because they are non-linear interactions, or more fundamentally, because models don't represent surface spatio-temporal heterogeneity (incl. phenology) and suffer from inaccurate boundary conditions in Africa where there are sparse observations?

We have revised this line in agreement with the reviewer (L329-331)

“Finally, interactions between poorly-observed mesoscale land heterogeneity and CI in the presence of directional shear may also contribute to the notable lack of skill...”

More broadly, the reviewer raises an interesting point which we cannot do justice to in the revised manuscript but include some thoughts here. The study we refer to (Vogel et al 2020) evaluates short-term ECMWF forecasts across the tropics, with West and Central Africa standing out as rainy regions where “even postprocessed [24-hour] forecasts are hardly better than climatology”. Elsewhere, forecast skill is much better. This points to errors in the initial conditions and/or the physics of convection. We would argue that some of the problem comes from systematic biases in SM-P coupling inherent in coarse resolution models that parameterise convection, as shown in Taylor et al (2012). Those models also struggle to represent organised convection, with or without a heterogeneous surface. In terms of land initial boundary conditions, a good NWP model ought to capture at least some of the spatial heterogeneity if it is assimilating SSM (e.g. both Met Office and ECMWF assimilate a coarse resolution version of ASCAT data). Those models also use screen level T and RH to nudge SM, and there is certainly a lack of those data in many parts of SSA. It may well be that the nature of the data assimilation system effectively coarsens the spatial resolution of the observations further. As for phenology, we agree that vegetation dynamics provides a source of predictability for the atmosphere. In a recent study (Taylor et al 2024), we showed that in West Africa, Vegetation Optical Depth (alongside SM and LST) retains a memory of past convective storm patterns out to 20+ days. However, during the rainy season, we detected an impact on the spatial distribution of subsequent convection pattern only out to 2-8 days, the timescale depending on how rain/water-limited the region is. That implies that in this region at least, convection responds most strongly to surface soil moisture from recent rainfall, rather than vegetation dynamics, which have time scales of several days to many weeks (Harris et al 2022). We would argue that the coupled land-atmosphere system in this region exhibits strong non-linear behaviour on time scales spanning multiple rain events, which limit predictability on sub-seasonal time scales.

Ln323 suggest “markedly stronger [when paired with a particular configuration of directional wind shear]”

Rewritten as (L334-335)

“markedly stronger when wind shear is accounted for”

Ln326 “within local dry soil” or “[over] dry soil”?

Changed (L337)

Ln327 reconsider use and definition of term “explosive”

We now introduce the term “explosive” at L116, which we think aptly captures the extremely rapid transition in the fastest-growing clouds.

‘Of these “explosive” events’

Ln331 suggest “models could [theoretically improve] early warning”

We have updated the text to emphasise the potential for improved early warning (L341-345).

“Our results indicate that incorporation of land surface state information into AI-based or Numerical Weather Prediction models has the potential to improve fine-scale prediction of thunderstorm initiation. This addresses a major limitation in the provision of early warnings of hazardous storms across the region”

Ln348 How is parallax corrected? This is the Methods section, afterall.

We have added the following (L360-361)

“using climatological profiles from ERA5 to convert cloud-top temperature to height.”

Ln359 “[on] a semi-empirical”

Changed (L372)

Ln361 suggest “[incremental] improvements”

Changed (L375)

Ln375 suggest “[vegetation root] access to groundwater”

Changed (L390)

Ln387 suggest “[precipitation] gauge”

Changed (L401)

Ln389 suggest “microwave [radar] are”

In fact the majority of microwave data used in IMERG are from passive sensors on multiple satellites, rather than the single radar-equipped GPM satellite.

Ln391-393 what fraction of total CIs analyzed in Fig 1 are analyzed for lightning?

The number of events is provided within each panel.

Ln400 why interpolate for 100-m winds rather than pick nearest neighbor?

We degrade the ERA5 resolution from 0.25 to 1 degree for the reasons set out in the Methods text (L413-415). To estimate the large-scale wind field at the initiation location, we found that the increased accuracy provided by interpolation (rather than nearest neighbour) slightly improves (i.e. sharpens) SM gradients.

Ln402 define angle ranges for each angle class

As noted above, we have improved the text at L416-418. We have also added a quadrant representation to the wind vectors in Figure 2 and Extended Data Figures.

Ln403 suggest “surface [SM heterogeneity], we”

We choose to retain “surface” rather than “SM” as the same approach is applied to LST (L418-419)

“on mesoscale surface heterogeneity”

Ln426 define subscript “sh”. Add definition of “favourable” and “unfavourable” to the Fig. 2 caption. Readers should not have to read the Methods to understand the main figures.

We defined subscript “sh” in the two preceding sentences (L438-441).

“A variant on ΔSM , ΔSM_{sh} , is defined using different box locations according to the direction of the mid-level flow relative to the low-level wind. These locations (plotted in Extended Data Fig. 5) are selected based on the composite-mean SM patterns shown in Figure 2a-d.”

On understanding Figure 2, we don’t think that readers need to consult the Methods section to gain a basic understanding as the main text introduces “favourable” and “unfavourable” SM contrasts (L178-180).

“To diagnose the impact of SM gradients on cloud growth, we define direction-specific “favourable” and “unfavourable” SM contrasts (ΔSM_{sh}), based on composite mean SM structures in Figures 2a-d (see Methods and Extended Data Fig. 6).”

Ln433-436 re: Results from...In summary, the study”. This is confusing that we are now discussing results from another study, or are they also used in this study. If so, specify more clearly where these results from another study are used/shown in this study and how it relates to New results (Figs 1-3) from this study.

To address this point, we have rewritten text at L450-451.

“In Figure 4, we place our results on SM-P feedbacks in a wider context by revisiting a previous global study of SM effects on afternoon rain. That analysis was based on...”

Ln443 “Sc” is undefined.

In fact, we introduce Sc in the main text (L257)

“as quantified by positive values of the cosine similarity of the low and mid-level wind vectors ($S_C(v_{low}, v_{mid})$)”

Ln444 “5deg [resolution]” and Ln445 “[CI] events.”

We have modified the text (L462-463) to read

“presenting data in Figure 4 when there are at least 50 contributing rain events within a 5° grid cell”

Ln473 re: “reduced sensitivity”. The authors have not calculated sensitivity. Is it correlation or correspondence?

We have retained the use of “sensitivity” here. The sensitivity of the LST-SM relationship to aerodynamic roughness can be computed directly from a modified Penman-Montieth equation, as shown for example in the cited study by Gallego-Elvira et al (2016).

Ln493 should the dataset DOIs be included instead?

We will follow editorial guidance on this point.

Ln504 it should be specified if the available code covers all analyses and figures, or a subset

We have modified the text (L518)

“Code for identifying and analysing CIs and the full CI dataset are available”

FIG.1.

Specify time period and source of data for SM, P, CI, elevation, and cloud cooling rate.

The reviewer is suggesting that we should include a lot of additional information that can already be found within the main text. We note that the guidelines for authors state “Each figure legend should begin with a brief title for the whole figure and continue with a short description of each panel and the symbols used.” On this specific point, we don’t consider it appropriate to add the details suggested.

Should it be specified these are “daytime-only” CIs

This is an important point, and one that we should have also mentioned in the article summary paragraph.

We have updated the text accordingly at L132

“Figure 1 Composite SM patterns preceding afternoon CI events”

And L16

“Analysing 2.2 million afternoon events”

Color bar should have a white increment (i.e., non-blue and non-red) for some +/- offset from 0. As is, a value of +0.00001 or -0.00001 would have red and blue shades, when they are very similar.

Is the color bar range truly [-1%, 1%]?

In the revised version we have used an alternative colour scale with white depicting SM anomalies within 0.2 of zero, and spanning a range from -2% to 2% (Figures 1, 2 and Extended Figures 6 and 7.)

Ln131 inappropriate use of “anomaly”? anomaly implies difference relative to climatology. should it be “difference”?

The two-dimensional field that we plot is indeed the SM anomaly from climatology, with the domain-mean value of that anomaly for each event removed (as described in the Methods section (L418-420)

“To focus on mesoscale surface heterogeneity, we remove the domain-mean of the (temporal) anomaly in SM and LSTA before compositing on a regular grid”.

Fig. why are the three dashed boxes superimposed on Fig.1a not also included in Fig. 1b?

The dashed boxes denote the composite locations used to compute Δ SM. We don’t compute Δ SM for the (much smaller sample of) extreme SM anomalies mapped in Figure 1b so don’t include the boxes in that panel.

Ln133 define “low-level”. 100-m.

Changed (L134)

Ln133 rephrase “upwards”. consider, “to top of page” or similar, so as not to be confused with positive vertical motion.

Good point. We have replaced “upwards” with “bottom-to-top” (L134-135).

Ln135 define “local” with a spatial scale length

The length scale is included in the definition of ΔSM in the main text (L107-108).

“We define ΔSM as the difference between the pixel just upstream of the CI and areas 50 km away (Figure 1).”

Fig 1c. why is the x-axis non-linear? and Ln136 re: “topographic standard deviation”. At what scale is this calculated? Is it a 200km radius of the primary CI location, or different?

We split the dataset into deciles of topographic standard deviation within a 30 km radius (as already noted in Methods (L431-434).

“For Figure 1c, all events with available SM data are grouped into deciles of the local (within 30 km) standard deviation in topographic height”.

We present the data on an x-axis of decile number rather than topographic standard deviation to more clearly demonstrate the gradual increase in ΔSM for each decile. The alternative approach leads to a more cluttered presentation of the data, with most of the distribution located very close together on the x-axis

Ln138 suggest “based on a [precipitation] threshold”

Changed (L139)

Ln138 re: dSM. Why is the solid box in Fig. 1a not dSM=0?

This comment made us realise that the variables plotted in the original figure were not sufficiently well-labelled, with a potential ambiguity that the y-axis label of Figure 1c (“dSM”) could refer to the colour bar. The colour bar in fact relates to the 2-dimensional SM anomaly field in panels a and b. To address this, we have adjusted the positioning of elements within Figure 1 and now label the colour bar (also repeated for Figure 2).

To answer the reviewer’s question, Figure 1a is not showing dSM, but the SM anomaly. dSM is the difference between the SM anomaly in the solid box compared to the mean over the three dashed boxes. To address this confusion, we have changed the nomenclature used throughout the manuscript to describe spatial differences in SM (and LST) anomalies from “dSM” and “dLST” to “ ΔSM ” and “ ΔLST ”.

Ln140 how is this 50-km standoff distance chosen. The choice is not justified in the Methods. Also, what is the dimension of these three boxes over which SM is spatially averaged?

We have added the following sentence in the methods (L430-431):

“We chose a distance of 50 km based on previous analysis highlighting SM gradients on length scales of 10-40 km (Taylor et al, 2011).”

That study showed that “variations in soil moisture on length scales of approximately 10–40 km exert a strong control on storm initiation”.

The reviewer has picked up an error in the original methods text – we stated that the three boxes were of width 50 km, rather than the 30 km we actually used.

FIG. 2.

Color bar label and units are missing.

As with Figure 1, we now include a label and units for the colour bar.

Ln156. It does not appear “As in Figure 1”, because the fields have not been rotated as they were in Fig. 1 (i.e., wind flow is not always towards the top of the page). Also, there are not three superimposed dashed boxes.

The original legend text was confusing. In fact, the SM anomaly data have been rotated exactly as in Figure 1 so the 100m wind is flowing bottom-to-top in panels a-e. The difference is that in Figure 2, the data from Figure 1a have now been split according to relative mid-level wind direction.

We have rewritten the offending sentence (L158-159) as

“Composite mean SM [shaded; %] around CI events, rotated according to the 100m wind, as in Figure 1.”

Ln157,162 “SM” or “dSM”?.

We have changed “SM” to “SM anomalies” in both cases.

Ln158-159 it should be clarified that there is a range of flow directions included in the composite, not just flow in the cardinal directions denoted by the vector plotted in the lower left corner of each panel plot

We have replaced the vector representation with a quadrant to address this point.

Ln159 is 3-hour accumulation window leading, lagging, or centered on CI event?

We have clarified (L161)

“3-hour post-initiation P accumulations”

Ln161 suggest “correlation coefficient between SM and P”

We have used $r(\text{SM}, \text{P})$ consistently throughout the text, and defined it at L192

Ln162 shear “class” or “configuration”?

Changed (L164).

Ln163 specify July 2024 to what end date (December 2024)?

Replaced with (L166)

“for July-December 2024”

Ln163 flashes over 3-hour period centered on CI event, or leading, or lagging?

Rewritten (L165)

“flashes over 3 hours starting 15 minutes prior to initiation”

Ln163 Cloud [top] cooling rate

Changed (L166)

Ln165 define “favourable” and favourable for what?

“Favourable” and “unfavourable” are defined in the main text. For clarity we have added “for CI” (L168).

Why is the line of symmetry in SM along the diagonal (100km downwind to 100km crosswind) in Fig. 2e?

Comparing Figure 2e with Figure 2d, it is important to remember that the number of events is 31 times smaller in the former. The composite mean patterns in Figure 2e are therefore noisier in general. There are several larger-scale differences between the two panels due to the sample size differences, including the presence of a larger scale left-to-right gradient in 2e, and some rotation in the pattern noted by the reviewer. However, the key feature (a dry ellipse centred around the initiation location) is clearly present in both. Similarly, the patterns in Extended Data Figure 6 (showing the other 3 configurations) are noisier compared to Figure 2a-c, but the local SM anomaly pattern emerges consistently in the smaller dataset coinciding with the availability of lightning data.

Note that we have also changed the colour of the lightning contour in panel e to better distinguish it from the black precipitation contours in Fig. 2 a-d.

As an aside, it would be interesting to investigate the diurnal cycle frequency for each low-mid level vertical wind shear configuration.

In Figure R2 we show the diurnal cycle of initiations for the 4 directional shear configurations. The diurnal cycle is very similar in all cases, with peak frequencies in early afternoon. With reverse shear (v650-) the diurnal cycle is slightly shifted to later in the afternoon. This might indicate a suppression of CI under conditions where shear-driven entrainment of drier air into the growing cloud is stronger. However, one should bear in mind that differences in the thermodynamic conditions sampled within the four groups may also contribute to differences in timing.

Figure R2 Contribution of CIs in a given hour to the total for the four directional shear configurations.

FIG. 3.

It is not entirely apparent that these simple schematics are fully-supported by the results shown in Fig. 2. Some representative cases drawn from the CI events analyzed may be more effective.

We believe the simple schematics do capture the key elements in Figure 2. Our analysis reveals that the location where CI occurs in the presence of SM heterogeneity is strongly related to the relative direction of the mid-level flow. The location of the warm cloud preceding initiation (green contours in Figure 2) tells us that the growing convection is moving with the mid-level wind. Based on numerical modelling simulations over 30-40 years, we know how SM heterogeneity influences the low-level wind field. In our observations, we find that CIs are favoured where the growing cloud can benefit from the enhanced cloud-relative moisture inflow provided by those SM-induced circulations. We also find that when the SM gradients in a direction and location that enhances the low-level circulation (and therefore cloud-relative inflow is especially strong), vertical cloud growth is significantly enhanced.

We have been careful to present Figure 3 and associated text as a proposed mechanistic interpretation (L202-203), based on what observations are available to us (SM, cloud-top temperature evolution). Of course, if we had observations of the 3-D wind and humidity fields we could be much more confident in that interpretation, based on real-world observations. In the future we plan to look at high resolution simulations of CI. That would provide a quantitative and much more complete picture of the key interactions between the low-level circulation and developing deep convection.

The reviewer wonders whether selecting representative cases might be a more effective way of summarising our findings than a schematic. Whilst we agree that real-world examples can be very helpful, it is always challenging to identify truly representative cases, and avoiding getting bogged down in describing case-specific details. In this case, we prefer to rely on our existing Figure 3.

Ln226 suggest “schematic of observationally-supported daytime soil moisture affect on convective initiation location under three vertical wind shear configurations” or similar. The phrase “soil moisture-controlled” should be removed.

We have replaced “controlled” with “affected” (L229). We choose not to refer to the 3 shear configurations in the title as that is only relevant for panel b.

In Fig3b legend, is it only “SM-driven circulation” or is it “terrain” or “surface flux gradient- or surface heating-driven circulation”?

The reviewer is correct that other drivers of low-level circulations could also influence CI in this manner, as we note in the discussion section (L310-312). However, we have limited our schematic to the driver which is the focus of our study, i.e. soil moisture.

FIG.4.

Ln260 is this dSM still defined as the difference between three box-averaged SM at 50-km distance (i.e., Fig.1) It is unclear.

Yes, this is the case. We hope that edits described above (and which appear earlier in the text) have clarified exactly what Δ SM is. We have used Δ SM consistently throughout the paper.

Ln268 suggest “shown as black [unfilled or hollow] bars” or similar.
Changed (L274)

Additional references of relevance to consider citing:

Ford, T. W., Steiner, J., Mason, B., & Quiring, S. M. (2023). Observation-driven characterization of soil moisture-precipitation interactions in the central United States. *Journal of Geophysical Research: Atmospheres*, 128, e2022JD037934. <https://doi.org/10.1029/2022JD037934>

Zhang, M. S., D. D. Turner, and D. Entekhabi, 2025: Relative Contributions of Surface Parcel and Atmospheric Environment to Convective Potential during Interstorm Periods. *J. Hydrometeor.*, 26, 1395–1406, <https://doi.org/10.1175/JHM-D-25-0045.1>.

Campbell, M. A., C. R. Ferguson, D. A. Burrows, M. Beauharnois, G. Xia, and L. F. Bosart, 2019: Diurnal Effects of Regional Soil Moisture Anomalies on the Great Plains Low-Level Jet. *Mon. Wea. Rev.*, 147, 4611–4631, <https://doi.org/10.1175/MWR-D-19-0135.1>.

Findell, K. L., and E. A. B. Eltahir (2003), Atmospheric controls on soil moisture-boundary layer interactions: Three-dimensional wind effects, *J. Geophys. Res.*, 108, 8385, doi:10.1029/2001JD001515, D8.

Thank you for these suggestions, including an interesting new study published after we submitted our manuscript! Unfortunately, there is a limit of 50 references in the journal so we are unable to cite all relevant literature.

References in our response

Barton, E. J. *et al.* Soil moisture gradients strengthen mesoscale convective systems by increasing wind shear. *Nature Geoscience* (2025). <https://doi.org/10.1038/s41561-025-01666-8>

Fletcher, J. K. *et al.* Tropical Africa’s first testbed for high-impact weather forecasting and nowcasting. *Bull. Am. Meteorol. Soc.* (2022). <https://doi.org/10.1175/bams-d-21-0156.1>

Findell, K. L. & Eltahir, E. A. B. Atmospheric controls on soil moisture-boundary layer interactions: Three-dimensional wind effects. *Journal of Geophysical Research: Atmospheres* **108** (2003). <https://doi.org/https://doi.org/10.1029/2001JD001515>

- Gallego-Elvira, B. *et al.* Global observational diagnosis of soil moisture control on the land surface energy balance. *Geophys. Res. Lett.* **43**, 2623-2631 (2016).
<https://doi.org/10.1002/2016gl068178>
- Guillod, B. P., Orlowsky, B., Miralles, D. G., Teuling, A. J. & Seneviratne, S. I. Reconciling spatial and temporal soil moisture effects on afternoon rainfall. *Nat Commun* **6** (2015).
<https://doi.org/10.1038/ncomms7443>
- Harris, B. L. *et al.* Satellite-Observed Vegetation Responses to Intraseasonal Precipitation Variability. *Geophys. Res. Lett.* **49**, e2022GL099635 (2022).
<https://doi.org/https://doi.org/10.1029/2022GL099635>
- Klein, C. & Taylor, C. M. Dry soils can intensify mesoscale convective systems. *Proceedings of the National Academy of Sciences* **117**, 21132-21137 (2020).
<https://doi.org/10.1073/pnas.2007998117>
- Koster, R. D. *et al.* Regions of strong coupling between soil moisture and precipitation. *Science* **305**, 1138-1140 (2004).
- LeBel, L. J. & Markowski, P. M. An Analysis of the Impact of Vertical Wind Shear on Convection Initiation Using Large-Eddy Simulations: Importance of Wake Entrainment. *Mon. Weather Rev.* **151**, 1667-1688 (2023). <https://doi.org/https://doi.org/10.1175/MWR-D-22-0176.1>
- Marquis, J. N., Varble, A. C., Robinson, P., Nelson, T. C., & Friedrich, K. (2021). Low-Level Mesoscale and Cloud-Scale Interactions Promoting Deep Convection Initiation. *Monthly Weather Review*, *149*(8), 2473-2495. <https://doi.org/https://doi.org/10.1175/MWR-D-20-0391.1>
- Morrison, H., Peters, J. M., Chandrakar, K. K., & Sherwood, S. C. (2022). Influences of Environmental Relative Humidity and Horizontal Scale of Subcloud Ascent on Deep Convective Initiation. *Journal Of The Atmospheric Sciences*, *79*(2), 337-359.
<https://doi.org/https://doi.org/10.1175/JAS-D-21-0056.1>
- Peters, J. M., Morrison, H., Nelson, T. C., Marquis, J. N., Mulholland, J. P., & Nowotarski, C. J. (2022a). The Influence of Shear on Deep Convection Initiation. Part I: Theory. *Journal Of The Atmospheric Sciences*, *79*(6), 1669-1690. <https://doi.org/https://doi.org/10.1175/JAS-D-21-0145.1>
- Peters, J. M., Morrison, H., Nelson, T. C., Marquis, J. N., Mulholland, J. P., & Nowotarski, C. J. (2022b). The Influence of Shear on Deep Convection Initiation. Part II: Simulations. *Journal Of The Atmospheric Sciences*, *79*(6), 1691-1711. <https://doi.org/https://doi.org/10.1175/JAS-D-21-0144.1>
- Taylor, C. M. *et al.* Frequency of Sahelian storm initiation enhanced over mesoscale soil-moisture patterns. *Nature Geosci* **4**, 430-433 (2011). <https://doi.org/10.1038/ngeo1173>
- Taylor, C. M. *et al.* Nowcasting tracks of severe convective storms in West Africa from observations of land surface state. *Environmental Research Letters* **17**, 034016 (2022).
<https://doi.org/10.1088/1748-9326/ac536d>

Taylor, C. M., Klein, C. & Harris, B. L. Multiday Soil Moisture Persistence and Atmospheric Predictability Resulting From Sahelian Mesoscale Convective Systems. *Geophys. Res. Lett.* **51**, e2024GL109709 (2024). <https://doi.org/10.1029/2024GL109709>

Vogel, P., Knippertz, P., Fink, A. H., Schlueter, A. & Gneiting, T. Skill of Global Raw and Postprocessed Ensemble Predictions of Rainfall in the Tropics. *Weather and Forecasting* **35**, 2367-2385 (2020). <https://doi.org/10.1175/waf-d-20-0082.1>

Response to Second Review from Referee#3

Below we respond to each comment from the referee, with their text shaded in grey.

This is a review of the revised manuscript titled, “Wind shear enhances soil moisture influence on rapid thunderstorm growth.” In general, the authors have done a fine job addressing the myriad of concerns raised by reviewers in the first round. Line numbers were omitted from the revision, however, which made writing this review more cumbersome.

I have two final major suggestions: 1) reconsider the title and 2) underscore uncertainty in the ERA5 winds used to compute shear.

We thank the reviewer for their suggestions on improving the final version of the manuscript. We apologise for the lack of line numbering in the previous submission. We included line numbers in our tracked changes version but inadvertently omitted them when creating a pdf of the manuscript itself.

With regard to the title, readers will find a disconnect between the title and the core thrust of analysis presented. The title reference to “rapid thunderstorm growth” is really pointing only to a single subpanel (i.e., Fig 2f) and Extended Fig. 6. The heart of the robust analysis is on convective initiation, not cloud cooling rate. Also, the title states that “wind shear enhances soil moisture influence.” The paper shows that wind shear can “modulate” or “mediate”, but not always “enhance”.

We appreciate the reviewer’s suggestions on the title, raising two distinct issues (“rapid *thunderstorm* growth” and “wind shear *enhances*”).

On the use of the term “thunderstorm”, we note that Figures 2e (not 2f) and Extended Fig. 6 show an average of 13.7-24.0 lightning flashes, localised around the CI location. Whilst we used rapidly-cooling cloud-top temperatures to create our CI dataset, this evidence from the 6-month period where we have lightning data demonstrates very clearly that these events typically produce a lot of lightning, therefore justifying our use of the word “thunderstorm”.

Our choice of the word “enhances” in the title is at the heart of the paper. Previous work has found that SM gradients influence CI, but here we show that the effect becomes stronger as wind shear increases. For example, when considering directional shear, the composite-mean SM gradients in Figures 2a-d are markedly sharper than the all-shear structure in Figure 1a. Figure 2f provides a clear demonstration that the SM influence on cloud cooling rate is enhanced as absolute wind shear increases. Here we combined cases within wind shear bins of 2 ms^{-1} and found that the cooling rates associated with favourable and unfavourable SM patterns diverge as the winds become more sheared. Moreover, Figure 3a provides an observationally-based mechanism for the SM effect on CI being enhanced as wind shear increases.

In sum, we consider that the title “Wind shear enhances soil moisture influence on rapid thunderstorm growth” is an accurate reflection of the work and prefer to retain it.

With regard to ERA5 winds, the authors wrote in their Response that they were “not aware of any such evaluation [of winds] due to the sparsity of wind observations (even at 10m) across the study area.” Using a dataset of unknown accuracy is a major limitation of this work. Given representativeness of ERA5 low-to-mid level wind shear is central to the authors’ conclusions,

they should consider disclosing its accuracy over sub-Saharan Africa is indeed a question mark. One way to do so, is to add a statement to the articles closing paragraph underscoring the need to verify low-level winds upon which the findings depend; improving forecasts of CI location is not only about additional incorporation of land surface state information to forecast systems.

This is a good suggestion by the reviewer, and we have added two sentences in the Discussion section to address it (L242-5).

“We note that our results depend on the realism of the large-scale wind shear in ERA5 over a relatively poorly-observed region of the world. Nonetheless, our analysis produces strongly contrasting pre-CI SM patterns when stratified into broad directional shear categories (Figure 2).”

We would however argue that this is not a *major* limitation. When we stratify our cases into 4 broad directional-shear categories, very clear distinctions emerge in antecedent SM patterns. Furthermore, we find the expected contrasts between these categories in the behaviour of observed (i.e. independent of ERA5) cloud and rainfall. In Figure 2a-d, pre-CI cloud growth (depicted by the green contours in each panel) occurs upstream (in a mid-level sense) from the CI location. Similarly maximum rainfall accumulations stretch out from the CI location in the direction of the mid-level wind. Finally, we note that broad differences in shear within our CI dataset are largely climatological, as evidenced in Figure 4c. Diversity in wind shear is primarily driven by well-understood large-scale circulation features such as the south-westerly West African Monsoon flow and the mid-level African Easterly Jet.

I also have a few minor comments:

1) Sensationalism. My suggestion is to reconsider the unnecessarily sensational term “explosive” in the manuscript.

A >55C/hr cooling rate is “explosive”, but not 50C/hr? It’s just a bit over dramatic.

We have replaced the term “explosive” with “extreme” in the summary (L15) and when introducing cloud growth rates (L114).

In another place, the authors use a sensational 68% figure, when they could instead explain the difference is ~2%: “the probability of a CI being classified as extreme is 68% higher for favourable”, in Fig. 2g, it looks like a much less remarkable difference of ~2% vs. ~4%.

We now explicitly mention the individual probabilities, as well as the relative increase (L153-5):

“For strong shear ($>12 \text{ ms}^{-1}$), the probability of a CI being classified as extreme (Figure 2g) is 5.17% for favourable SM configurations, compared to 3.08% for unfavourable SM, a relative increase of 68%.”

2) The second sentence of the Discussion could be misleading. As written, the dependence of “SM patterns strongly influence the vertical growth of storm clouds” (Fig 2f) and “favorable SM patterns producing more lightning and rainfall” (Figs 2d,e,g) on coincident favorable wind shear conditions is not strictly clear. Suggest rephrasing.

We have removed the final part of this sentence to avoid confusion. It now reads (L236-9)

“It reveals for the first time the important mediating effect of wind shear on SM-CI relationships, with SM patterns strongly influencing the vertical growth of storm clouds.”

3) Later in the Discussion, “Favourably aligned [] SM-induced circulations...These are cases for which there is little indication of the imminent storm from satellite imagery even an hour ahead...” As written, this suggests the statement only holds for favourable SM cases, when it likely holds for some unfavourable SM cases, too. Suggest rewording.

This is a good point. We have rewritten the text to avoid that suggestion (L276-7):

“Such conditions contribute disproportionately to cases for which there is little indication...”

4) Authors should double check references to Extended Figures cite the correct Extended Figure number. Also, intended figure references may be missing from the statement, “We use the same approach to create composite maps based on cloud-top temperature (at 5km) and lightning (at 10km).” Above that line, there is a small inaccuracy where it states “We degrade...”, when really ECMWF served the data at the degraded resolution.

We have checked all citations to Extended Data Figures and did not find any errors there. We did note an error in the previous “response to reviewers” document however, which likely prompted this comment.

We have amended the statement about composite maps (L558-60)...

“We use the same approach to create composite maps based on cloud-top temperature (at 5km) and lightning (at 10km) (Figure 2 and Extended Data Figure 6).”

We have tweaked the wording on our choice of resolution of ERA-5 wind data as ECMWF did indeed downgrade the resolution rather than us (L546-7).

“We prefer 1° resolution to the default (0.25°) to emphasise...”

Response to Second Review from Referee#3

Below we respond to each comment from the referee, with their text shaded in grey.

This is a review of the revised manuscript titled, “Wind shear enhances soil moisture influence on rapid thunderstorm growth.” In general, the authors have done a fine job addressing the myriad of concerns raised by reviewers in the first round. Line numbers were omitted from the revision, however, which made writing this review more cumbersome.

I have two final major suggestions: 1) reconsider the title and 2) underscore uncertainty in the ERA5 winds used to compute shear.

We thank the reviewer for their suggestions on improving the final version of the manuscript. We apologise for the lack of line numbering in the previous submission. We included line numbers in our tracked changes version but inadvertently omitted them when creating a pdf of the manuscript itself.

With regard to the title, readers will find a disconnect between the title and the core thrust of analysis presented. The title reference to “rapid thunderstorm growth” is really pointing only to a single subpanel (i.e., Fig 2f) and Extended Fig. 6. The heart of the robust analysis is on convective initiation, not cloud cooling rate. Also, the title states that “wind shear enhances soil moisture influence.” The paper shows that wind shear can “modulate” or “mediate”, but not always “enhance”.

We appreciate the reviewer’s suggestions on the title, raising two distinct issues (“rapid thunderstorm growth” and “wind shear *enhances*”).

On the use of the term “thunderstorm”, we note that Figures 2e (not 2f) and Extended Fig. 6 show an average of 13.7-24.0 lightning flashes, localised around the CI location. Whilst we used rapidly-cooling cloud-top temperatures to create our CI dataset, this evidence from the 6-month period where we have lightning data demonstrates very clearly that these events typically produce a lot of lightning, therefore justifying our use of the word “thunderstorm”.

Our choice of the word “enhances” in the title is at the heart of the paper. Previous work has found that SM gradients influence CI, but here we show that the effect becomes stronger as wind shear increases. For example, when considering directional shear, the composite-mean SM gradients in Figures 2a-d are markedly sharper than the all-shear structure in Figure 1a. Figure 2f provides a clear demonstration that the SM influence on cloud cooling rate is enhanced as absolute wind shear increases. Here we combined cases within wind shear bins of 2 ms^{-1} and found that the cooling rates associated with favourable and unfavourable SM patterns diverge as the winds become more sheared. Moreover, Figure 3a provides an observationally-based mechanism for the SM effect on CI being enhanced as wind shear increases.

In sum, we consider that the title “Wind shear enhances soil moisture influence on rapid thunderstorm growth” is an accurate reflection of the work and prefer to retain it.

With regard to ERA5 winds, the authors wrote in their Response that they were “not aware of any such evaluation [of winds] due to the sparsity of wind observations (even at 10m) across the study area.” Using a dataset of unknown accuracy is a major limitation of this work. Given representativeness of ERA5 low-to-mid level wind shear is central to the authors’ conclusions, they should consider disclosing its accuracy over sub-Saharan Africa is indeed a question mark. One way to do so, is to add a statement to the articles closing paragraph

underscoring the need to verify low-level winds upon which the findings depend; improving forecasts of CI location is not only about additional incorporation of land surface state information to forecast systems.

This is a good suggestion by the reviewer, and we have added two sentences in the Discussion section to address it (L242-5).

“We note that our results depend on the realism of the large-scale wind shear in ERA5 over a relatively poorly-observed region of the world. Nonetheless, our analysis produces strongly contrasting pre-CI SM patterns when stratified into broad directional shear categories (Figure 2).”

We would however argue that this is not a *major* limitation. When we stratify our cases into 4 broad directional-shear categories, very clear distinctions emerge in antecedent SM patterns. Furthermore, we find the expected contrasts between these categories in the behaviour of observed (i.e. independent of ERA5) cloud and rainfall. In Figure 2a-d, pre-CI cloud growth (depicted by the green contours in each panel) occurs upstream (in a mid-level sense) from the CI location. Similarly maximum rainfall accumulations stretch out from the CI location in the direction of the mid-level wind. Finally, we note that broad differences in shear within our CI dataset are largely climatological, as evidenced in Figure 4c. Diversity in wind shear is primarily driven by well-understood large-scale circulation features such as the south-westerly West African Monsoon flow and the mid-level African Easterly Jet.

I also have a few minor comments:

1) Sensationalism. My suggestion is to reconsider the unnecessarily sensational term “explosive” in the manuscript.

A >55C/hr cooling rate is “explosive”, but not 50C/hr? It’s just a bit over dramatic.

We have replaced the term “explosive” with “extreme” in the summary (L15) and when introducing cloud growth rates (L114).

In another place, the authors use a sensational 68% figure, when they could instead explain the difference is ~2%: “the probability of a CI being classified as extreme is 68% higher for favourable”, in Fig. 2g, it looks like a much less remarkable difference of ~2% vs. ~4%.

We now explicitly mention the individual probabilities, as well as the relative increase (L153-5):

“For strong shear ($>12 \text{ ms}^{-1}$), the probability of a CI being classified as extreme (Figure 2g) is 5.17% for favourable SM configurations, compared to 3.08% for unfavourable SM, a relative increase of 68%.”

2) The second sentence of the Discussion could be misleading. As written, the dependence of “SM patterns strongly influence the vertical growth of storm clouds” (Fig 2f) and “favorable SM patterns producing more lightning and rainfall” (Figs 2d,e,g) on coincident favorable wind shear conditions is not strictly clear. Suggest rephrasing.

We have removed the final part of this sentence to avoid confusion. It now reads (L236-9)

“It reveals for the first time the important mediating effect of wind shear on SM-CI relationships, with SM patterns strongly influencing the vertical growth of storm clouds.”

3) Later in the Discussion, “Favourably aligned [] SM-induced circulations... These are cases for which there is little indication of the imminent storm from satellite imagery even an hour ahead...” As written, this suggests the statement only holds for favourable SM cases, when it likely holds for some unfavourable SM cases, too. Suggest rewording.

This is a good point. We have rewritten the text to avoid that suggestion (L276-7):

“Such conditions contribute disproportionately to cases for which there is little indication...”

4) Authors should double check references to Extended Figures cite the correct Extended Figure number. Also, intended figure references may be missing from the statement, “We use the same approach to create composite maps based on cloud-top temperature (at 5km) and lightning (at 10km).” Above that line, there is a small inaccuracy where it states “We degrade...”, when really ECMWF served the data at the degraded resolution.

We have checked all citations to Extended Data Figures and did not find any errors there. We did note an error in the previous “response to reviewers” document however, which likely prompted this comment.

We have amended the statement about composite maps (L558-60)...

“We use the same approach to create composite maps based on cloud-top temperature (at 5km) and lightning (at 10km) (Figure 2 and Extended Data Figure 6).”

We have tweaked the wording on our choice of resolution of ERA-5 wind data as ECMWF did indeed downgrade the resolution rather than us (L546-7).

“We prefer 1° resolution to the default (0.25°) to emphasise...”